# Subjective valuation as a domain-general process in creative thinking

Gino Battistello [1] ✉, Sarah Moreno-Rodriguez [1], Emmanuelle Volle [1,2] & Alizée Lopez-Persem [1,2] ✉

Is a talented painter also a proficient writer? The ongoing discourse on whether creativity operates through domain-general or domain-specific mechanisms has led to challenges in our understanding of the creative process. Prior research suggests that creativity comprises two phases: idea generation and evaluation. A recent framework has proposed that the evaluation phase involves a valuation process which occurs upstream of the selection of an idea. In this framework, the value assigned to an idea, i.e., how much one likes an idea, energizes its production and drives its selection. While the role of valuation has been demonstrated in verbal semantic creativity, its domain generality remains to be tested. In this study, we assessed whether valuation is a domain-general or domain-specific process. Seventy-three participants engaged in three creativity tasks (producing semantic associations, alternate object uses, and drawings) followed by rating tasks. Using computational modeling, we found that a consistent valuation mechanism governs idea valuation across different domains. Specifically, the same value function and value parameters were shared across the evaluation of word associations, object uses and drawing completions. These findings advance our understanding of the evaluation phase of creativity, portraying the valuation component as inherently domain-general. Identifying such core components of creative ideation contributes to elucidating the cognitive mechanisms underlying creativity and provides empirical support for including valuation as a core process in creativity.

Creativity is defined as the ability to produce ideas or objects that are both original and adequate to the context[1,2]. Creativity can be expressed in various domains (e.g., art, literature, science, etc.). The study of creativity as a cross-domain process has fueled discussions in the field, giving rise to theories that support either domain-specific[3] or domain-general[4] definitions of creativity. Nevertheless, several theories provide a nuanced understanding, proposing that creativity involves both general and specific processes[5–7]. This view is supported by results in multi-domain creativity questionnaires. For instance, it has been shown that scores on creative activities and achievements are correlated across various domains (e.g., literature, music, art, etc., see ref. 8 for an exhaustive list), supporting the domain-generality argument. However, stronger correlations are observed between similar domains compared to dissimilar ones, supporting the domain-specificity argument[8]. For instance, scores of creative achievements and activities in the literature domain tend to correlate with both achievements and activities in a wide range of domains, such as culinary and visual arts, but the highest correlation remains for creative writing itself. These findings from questionnaires are consistent with empirical evidence[9] indicating that life experiences, such as education and cultural background, have a stronger impact on domain-

specific creativity than on domain-general creativity. Overall, these findings suggest that creativity has both domain-general and domain-specific mechanisms, the latter potentially developed from previous experiences and domain expertise[10,11]. A key remaining question is the extent to which the core cognitive mechanisms underlying creative thinking are domain-general or domain-specific.

Previous frameworks have proposed that creative thinking relies on two complementary mechanisms: generation, comprising spontaneous and goal-directed processes that generate remote associations, and evaluation, encompassing controlled processes, such as monitoring, inhibition, and selection of a response[1,12]. Currently, there is limited experimental data to determine whether each of these processes is domain-general or domain-specific. Brain imaging studies, which aim to identify the neural correlates of creativity, have provided mixed results. Many studies have identified two brain networks engaged in creativity tasks, highlighting their consistency across various domains (for reviews, see refs. 13,14). Concretely, it has been found that the Default Mode Network (DMN) is associated with idea generation, while the Executive Control Network (ECN) is linked to evaluative and controlled processes[13–18] in multiple domains. However, direct

[1]FrontLab, Sorbonne University, Institut du Cerveau - Paris Brain Institute - ICM, Inserm, CNRS, AP-HP, Hôpital de la Pitié Salpêtrière, Paris, France. [2]These authors contributed equally: Emmanuelle Volle, Alizée Lopez-Persem. ✉e-mail: gn.battistello@gmail.com; lopez.alizee@gmail.com

comparisons between domains have revealed both shared and distinct patterns of brain connectivity within these networks[19]. Furthermore, meta-analyses of fMRI data comparing different domains have identified brain regions with both domain-general and domain-specific involvement in creativity[15,20]. Whether domain-specific regions support distinct core creativity processes beyond other processes such as low-level information processing or response production remains an open question. To characterize if creativity operates in a general or a specific manner across domains, it is crucial to delve into the underlying mechanisms. Therefore, this study focuses on the evaluation stage, probing whether its valuation subprocess differs between domains or exhibits generality.

Our previous work[21] has provided a framework that allows to dissociate evaluative processes from generative processes during a creativity task and decomposes evaluation into three specific subprocesses: (i) the monitoring of ideas' originality and their adequacy to the context, (ii) the valuation of the ideas, i.e. the assignment of a subjective value based on originality and adequacy, according to individual preferences, and (iii) the selection of the idea with the highest subjective value. This framework was tested in an experimental design that combined a semantic creativity task and rating tasks of participants' responses in terms of originality, adequacy, and subjective values. Model fitting and simulations showed that subjective value drove the selection of an idea among potential candidates, in three main findings. First, we found that, during the creativity task, preferred ideas were provided faster than less valued ideas. These results suggest that the likeability of an idea energizes its production, emphasizing the potential underlying motivational role of valuation in creative idea production[22]. Indeed, valuation leads to behavioral energization[23], meaning that subjective values predict how much effort is put into an action, or in our case, how fast an action is performed, reflecting motivation. Second, computational modeling[21] revealed that an idea's subjective value relies on a combination of its adequacy and originality. This result aligns with the view that adequacy and originality are critical dimensions of creativity, consistent with the product-based definition of creativity, and previously proposed as the guiding factors of idea selection[24]. Third, we observed individual differences in how adequacy and originality contribute to subjective value, and, interestingly, individuals favoring originality tended to be more creative (as measured by a battery of creativity tests and questionnaires). This indicates that individual preferences play a role in creativity. Notably, these three main behavioral results were replicated in a recent study that examined their underlying neural correlates[25]. In the present study, we aim to generalize these results to other domains of creativity. We hypothesize that the valuation (as a subprocess of evaluation) is domain-general and that individual preferences are stable across domains. These hypotheses are supported by previous findings in the cognitive neuroscience of value-based decision-making which have demonstrated that preferences are stable across tasks[26], types of effort[27], and variations in experimental design (e.g., risk-seeking behavior is consistent for both real and hypothetical options)[28].

To test our main hypotheses, we used a similar experimental design as in prior studies[21,25], i.e., by combining creativity and rating tasks. As previously, we used the Free Generation of Associates Task (previously called FGAT, and here simplified to Word task): a simple word-to-word generation task reflecting the ability to produce remote associations either spontaneously (with the instruction to give the first word that comes to mind; FGAT-First condition) or intentionally (with the instruction to think creatively; FGAT-Creative condition). We supplemented this experimental design with two additional Free Generation tasks: a drawing task to target figural creativity, and an object-uses task, inspired by the Alternate Uses Task (AUT), classically used to assess creative abilities, and a gold standard method to investigate divergent thinking. These two tasks were also split into a First and a Creative condition, followed by likeability, adequacy, and originality ratings.

Our specific hypotheses were that (i) creative idea production is energized by valuation in the three creative domains (supporting the domain-general role of motivation), (ii) subjective values are built on both the adequacy and originality of response, and similarly across the three

domains, and (iii) individual differences in preferences are related to creative abilities in real-life.

## Methods

### Participants
Seventy-five participants (self-reported gender indicated 52 women, 21 men, and 2 non-binary participants) were recruited via an online advertisement distributed through the RISC (Relais d'Information en Sciences Cognitives) and the Paris Brain Institute mailing lists. Two participants were excluded due to a misunderstanding of the instructions, resulting in a final sample of 73 participants (50 women, 21 men, and 2 identifying as non-binary; $M_{Age} = 27.5 \pm 4.7$ SEM). No data was collected on race or ethnicity. All participants were tested at the PRISME platform of the Paris Brain Institute. Inclusion criteria required participants to be 20 to 40 years old, being a native French speaker, and have no severe auditory or visual impairment (corrected-to-normal vision was accepted). Volunteers were excluded if they reported neurological deficits, affective disorders, or were on current psychiatric medication. Additionally, participants were instructed to avoid psychotropic substance consumption the week preceding the experiment and refrain from alcohol consumption the night before the experiment. The initial sample size was determined to ensure sufficient statistical power for detecting interindividual correlations between participants' model parameters and their scores from creativity assessment questionnaires. The project was approved by an ethics committee (CPP OUEST II - ANGERS, RCB: 2019-A02511-56, SI/CPP: 19.01012.201987). Participants provided informed consent and received compensation for their participation, amounting to 75 euros by bank transfer.

### Preregistration
This study was not preregistered before data collection and analysis.

### Experimental design
All the tasks for the experiment were designed on MATLAB (R2021a) with Psychtoolbox[29] (http://psychtoolbox.org/) and ran on MATLAB (R2020b).

**Free Generation tasks.** Participants performed three Free Generation tasks (words associations, object uses, and abstract drawings). Each task comprised two experimental conditions: the control condition (First), in which participants provided the first response that came to mind given the cue, and the creative condition (Creative), in which participants were instructed to think creatively and provide an unusual, creative association in response to the same cues.

**Free Generation of Words (Words task).** This task, also known as the Free Generation of Associates Task (FGAT)[21,30] involved presenting participants with single cue words. Participants had to propose associated words as responses, forming word-word associations. For instance, given the cue "mother", responses could be "learning" or "nature". Thirty cue words were selected from the Dictaverf dictionary of associations (http://dictaverf.nsu.ru/)[31].

**Free Generation of Alternate Uses (Uses task).** Inspired from the AUT[32,33], objects' names were presented and participants had to propose a common use in the First condition, and an alternate use in the Creative condition to form object-use associations. For instance, given the cue "knife", the response could be "to cut something" in the First condition and "as a mirror" in the Creative condition. Thirty names of objects were selected from previous studies[34,35].

**Free Generation of Drawings (Drawings task).** Inspired by the Torrance subtests[36] and the work of Barbot et al.[37], this task involved presenting abstract shapes to participants, who were instructed to complete them, either spontaneously (First condition) or creatively (Creative condition), to form shape-drawing associations. Thirty abstract shapes, or droodles, were extracted from the study of Nishimoto et al.[38].

All cues are available in Supplementary Method 1.

Procedure. Each participant completed the three Free Generation tasks (Words, Uses, Drawings) in a pseudorandom order, with the First condition always preceding the Creative condition. Each condition consisted of 3 training trials and 30 test trials (adding up to 180 trials per participant across the three tasks and two conditions). In the First condition, each trial began with a fixation cross displayed for 1 s, followed by the presentation of a cue on the screen. In the Words and Uses domains, participants had up to 10 s to press the space bar of the keyboard to indicate that they had a response in mind. If participants did not respond within this time limit, the next fixation cross appeared, preceded by a message indicated that the trial is missed. If they pressed the space bar within the allotted time, they were given 30 s to type their response using the keyboard. Participants provided their responses using the keyboard. After providing their response, participants validated it by pressing the return key, which initiated the next trial. For the Drawings task, as soon as the cue (i.e., abstract shape) appeared on the screen, participants could create a figurative drawing using the computer mouse. They had 30 s to complete each drawing. Erasing was not allowed to capture the first idea that came to mind in the First condition, encourage participants to think before starting to draw in the Creative condition, and to simplify the computation of drawing speed. Participants were informed before the task that erasing was not allowed. Participants could press the return key to submit their drawing before the time limit. If the time limit expired, the drawing was automatically saved, and the next trial began.

In the Creative conditions, the procedure was the same, except that participants had 20 s to press the space bar in the Words and Uses domains and 60 s to complete their drawings in the Drawings domain.

Rating tasks. During the rating tasks, participants rated their associations across the three domains (Words, Uses, and Drawings) from the three Free Generation tasks previously completed (i.e., their word-word, object-use, shape-drawing associations), as well as additional associations pre-designed before the experiment. Participants first completed the likeability rating tasks for each domain, followed by the adequacy and originality rating tasks. The domains were blocked and completed in pseudo-random order. The same set of associations was presented across all rating tasks. Participants rated each association by moving a cursor using the left and right arrow keys along a horizontal scale (ranging from 0 as the lowest to 100 as the highest value). Once satisfied with their rating, they pressed the return key to validate their response. Each rating task included 3 training trials and, on average, 120 test trials, resulting in a total of around 1080 ratings per participant across all tasks and domains.

The number of trials for the Uses and Drawings rating tasks was fixed at 120 trials, consisting of the associations produced by the participants (i.e., 30 from First and 30 from Creative conditions) and associations from a pre-defined database (i.e., 20 First, 20 Creative, and 20 unrelated associations). For the Words task, an online spell check was applied during the Free Generation task to ensure the validity of responses. All valid responses from the First and Creative conditions were included in the rating tasks. To ensure consistency, additional responses from the Dictaverf database were included to maintain an equal number of First (high and low associative frequency), Creative (high and low associative frequency), and unrelated associations (see Supplementary Method 2 for more details on the selection of associations).

Likeability ratings. During the likeability rating tasks, participants were asked to picture themselves in the Creative condition of the Free Generation tasks and evaluate how much they liked or were satisfied with their response. If the association was not their own, they were asked to rate how much they would have liked providing the proposed association.

Adequacy and originality ratings. Once the likeability ratings for all domains were completed, the associations were presented again for adequacy and originality ratings. Participants were instructed to evaluate how appropriate and how original they found each association. The order of adequacy and originality ratings was randomized across trials to prevent order effects.

**Creativity assessment.** After completing the rating tasks, participants were asked to fill out a set of questionnaires composed of the ICAA[8] and a self-report of their perceived creativity level. This session was conducted using Qualtrics, an online survey platform, and was self-paced. The Inventory of Creative Activities and Achievements (ICAA) provides an ecological evaluation of creative behavior by assessing real-life creative activities and achievements across eight domains (literature, music, crafts, cooking, sports, visual arts, performing arts, and science and engineering). For each domain, participants were asked to indicate how frequently, over the past ten years, they had engaged in activities and achieved milestones specific to that domain. For each participant, we computed the C-Act (i.e., activities) and C-Ach (i.e., achievements) scores, respectively, as the mean score of creative activities and achievements across all domains. In addition to the ICAA, participants completed a self-report in which they indicated their perceived level (i.e., Do you think you are a creative person?) of creativity on a scale from 1 (non-creative at all) to 100 (highly creative). This self-report provided an assessment of participants' perception of their creativity.

### Statistical analysis
Analyses were performed on MATLAB (R2021a). The computation of AuDra and SemDis scores was done on Python (respectively 3.1 and 3.9). Data distribution was assumed to be normal, but this was not formally tested.

**Behavioral measures in generation and rating tasks.** The behavioral measures in Free Generation tasks were creativity scores, response time, and production speed. In the rating tasks, the main measures were the likeability, adequacy, and originality ratings. If not specified otherwise, linear regressions were conducted at the participant level using normalized variables. Significance was tested at the group level using a one-sample, two-tailed t-test on regression coefficient estimates.

**Automatic scoring of response's creativeness.** Participants' responses were scored using three different deep-learning pre-trained models. To assess the semantic proximity between the cue word and the participant's responses in the Words domain, and use semantic distance as a proxy of creativeness, we used word2vec cosine similarity[39]. Word2vec is a word embedding algorithm based on neural networks that learns vector representations of the words in a text, such that words with similar contexts are represented by numerically close vectors[39]. We downloaded a pre-trained word2vec model for French from https://fauconnier.github.io/ ref. [40]. This model was built from a 1.6-billion-word corpus sourced from websites with .fr domains, as described in Baroni et al.[41]. The cosine similarity ranges between −1 (low similarity) and 1 (high similarity), indicating the semantic similarity between two words. For each participant's response in the Words domain, we calculated semantic distance as the negative cosine similarity between the response and the cue.

In addition to word2vec, recent models allowed us to extract creativity scores from drawings (AuDra)[42] and objects' uses (SemDis)[43]. AuDra is based on Res-Net-18, a Convolutional Neural Network (CNN) trained on a large dataset of images (ImageNet) and adapted with a top layer trained on images associated with human creativity ratings. AuDra extracts the features of each image (i.e., drawings), and assigns a creativity score. SemDis is a model based on BERT (Bidirectional Encoder Representations from Transformers) and computes the Maximum Average Distance (MAD) for each response. This represents the maximum semantic distance between the cue (i.e., name of an object) and a token in the response (i.e., an element in the sentence). Following the authors' recommendation, we used the multilingual version of BERT to extract features and compute the MAD between the cues and responses, all written in French. For the Uses domain, we conducted control analyses using a recent alternative method for automatic scoring (Open Creativity Scoring with Artificial Intelligence, OSCAI)[44]. All results

presented with SemDis scores were replicated (including non-significance) with OSCAI scores (Supplementary Note 3 and Supplementary Fig. 4). Additionally, we validated the reliability of AuDra scores by comparing them with human expert ratings. We found good reliability for scoring drawings with AuDra (Supplementary Note 4). Overall, both AuDra and SemDis provide automated scoring of the creativeness of participants' responses. To assess the mean differences in creativity scores between the First and Creative conditions across the three domains, we performed one-sample, two-tailed t-tests.

**Response time and production speed.** In the Free Generation tasks, participants provided single words (Words domain), short sentences describing the use of an object (Uses domain), or drawings (Drawings domain) as responses. The response time (RT) was computed as the difference between the cue onset time and the space bar press for the Words and Uses domains, or the first mouse click for the Drawings domain. The space bar pressing (or first click) marked the moment when the participant had an idea in mind and was ready to provide it, capturing the time taken to generate a response. We also computed the production speed (PS) as the number of characters written (for Words and Uses) or pixels drawn (for Drawings) over the production time. Production time was defined as the interval between the first and the last character written or pixel drawn.

In addition to behavioral measures (e.g., RT, PS for all domains, and drawing time (DT) and length (DL) for the Drawings domain), each participant's response in Free Generation tasks was associated with a rating of likeability, originality, and adequacy. To examine the energization of behavior, we regressed behavioral measures independently against likeability ratings. These regressions were conducted at the individual level using normalized variables. The significance of the regression coefficient was then assessed at the group level using one-sample two-tailed t-tests against zero. Note that one participant provided a rating of 100 for all their responses from the Creative condition in the Uses domain. We thus excluded this participant from the analyses of RT and PS in the Uses domain.

**Model fitting and comparisons on data from the rating tasks.** The models were fitted to the data at the individual level using the Matlab VBA toolbox (available at http://mbb-team.github.io/VBA-toolbox/), which implements Variational Bayesian Analysis under the Laplace approximation[45,46]. This iterative algorithm provides a free-energy approximation to the marginal likelihood or model evidence, which represents a natural trade-off between model accuracy (goodness of fit) and complexity (degrees of freedom)[47,48]. Additionally, the algorithm estimates the posterior density over the model-free parameters, starting with Gaussian priors (i.e., defined by a Gaussian distribution with a mean and a variance). Individual log-model evidence was then taken to the group-level random-effect Bayesian model selection (RFX-BMS) procedure[45,49]. RFX-BMS provides an exceedance probability (Xp) that measures how likely it is that a given model (or family of models) is more frequently implemented relative to all the others considered in the model space in the population from which participants are drawn.

Assessing value functions across the three domains. To investigate how originality (O) and adequacy (A) are integrated into a subjective value, we used the following 12 value functions capturing linearly (or not) the relationship between likeability (L), adequacy (A), and originality (O):

• **Linear**

$$L_i = \beta A_i \tag{1}$$

$$L_i = \alpha O_i + (1 - \alpha)A_i \tag{2}$$

$$L_i = \alpha O_i + \beta A_i \tag{3}$$

• **Linear with an interaction term**

$$L_i = \gamma O_i * A_i \tag{4}$$

$$L_i = \alpha O_i + (1 - \alpha)A_i + \gamma O_i * A_i \tag{5}$$

$$L_i = \alpha O_i + \beta A_i + \gamma O_i * A_i \tag{6}$$

• **Non-linear (with the same non-linearity on both dimensions)**

$$L_i = \left(\alpha O_i^\delta + \beta A_i^\delta\right)^{\frac{1}{\delta}} \tag{7}$$

$$L_i = \left(\alpha O_i^\delta + (1 - \alpha)A_i^\delta\right)^{\frac{1}{\delta}} \text{ (CES)} \tag{8}$$

$$L_i = \alpha O_i^\delta + \beta A_i^\delta \tag{9}$$

• **Non-linear (with different non-linearity on both dimensions)**

$$L_i = \beta A_i^\delta \tag{10}$$

$$L_i = \alpha O_i^\delta + (1 - \alpha)A_i^\varepsilon \tag{11}$$

$$L_i = \alpha O_i^\delta + \beta A_i^\varepsilon \tag{12}$$

Greek letters correspond to free parameters estimated with the fitting procedure described above; i refers to a given cue-response association. Note that the α parameters are constrained between 0 and 1 for value functions (2), (5), (8), and (11) during the fitting procedure.

All possible combinations of value functions were tested across domains. In other words, we tested whether each domain could be explained by a specific value function or whether a single value function could account for likeability ratings across all three domains. This resulted in $12^3$, so 1718 models, that were grouped into two families: models where the same value function explains likeability across all domains and models where different value functions explain likeability across domains.

Assessing the value parameters across the three domains. According to the results from the previous analysis, the Constant Elasticity of Substitution (CES, value function 8) was identified as the value function providing the best account to explain participants' behavior across the three domains. Therefore, the subsequent analysis focused on this value function (Eq. 13). To assess whether the same parameters of the CES captured likeability ratings across all three domains, we conducted a model comparison over the model space derived from the following full model:

$$
\begin{bmatrix} L_{\text{Words}} \\ L_{\text{Uses}} \\ L_{\text{Drawings}} \end{bmatrix} = \begin{bmatrix} \text{CES}_{\text{Words}}\left(A_{\text{Words}}, O_{\text{Words}}; \begin{cases} \alpha_{\text{Words}} \\ \delta_{\text{Words}} \end{cases}\right) \\ \text{CES}_{\text{Uses}}\left(A_{\text{Uses}}, O_{\text{Uses}}; \begin{cases} \alpha_{\text{Uses}} = \alpha_{\text{Words}} + \alpha'_{\text{Uses}} \\ \delta_{\text{Uses}} = \delta_{\text{Words}} + \delta'_{\text{Uses}} \end{cases}\right) \\ \text{CES}_{\text{Drawings}}\left(A_{\text{Drawings}}, O_{\text{Drawings}}; \begin{cases} \alpha_{\text{Drawings}} = \alpha_{\text{Words}} + \alpha'_{\text{Drawings}} \\ \delta_{\text{Drawings}} = \delta_{\text{Words}} + \delta'_{\text{Drawings}} \end{cases}\right) \end{bmatrix} \tag{13}
$$

with six free parameters: $\alpha_{\text{Words}}$, $\alpha'_{\text{Uses}}$, $\alpha'_{\text{Drawings}}$ and $\delta_{\text{Words}}$, $\delta'_{\text{Uses}}$, $\delta'_{\text{Drawings}}$. The full model has six inputs (adequacy and originality ratings from each domain) and three outputs (likeability ratings from each domain). In this equation, α' and δ' are additive parameters. If their prior variance and prior mean are set to zero, the model has only two free parameters, corresponding to our main hypothesis: parameters are stable across domains (reduced model called all equal "All ="). All intermediate models (such as only two domains with shared parameters) were tested. Individual log-model evidence for all the models was computed using the Savage-Dickey

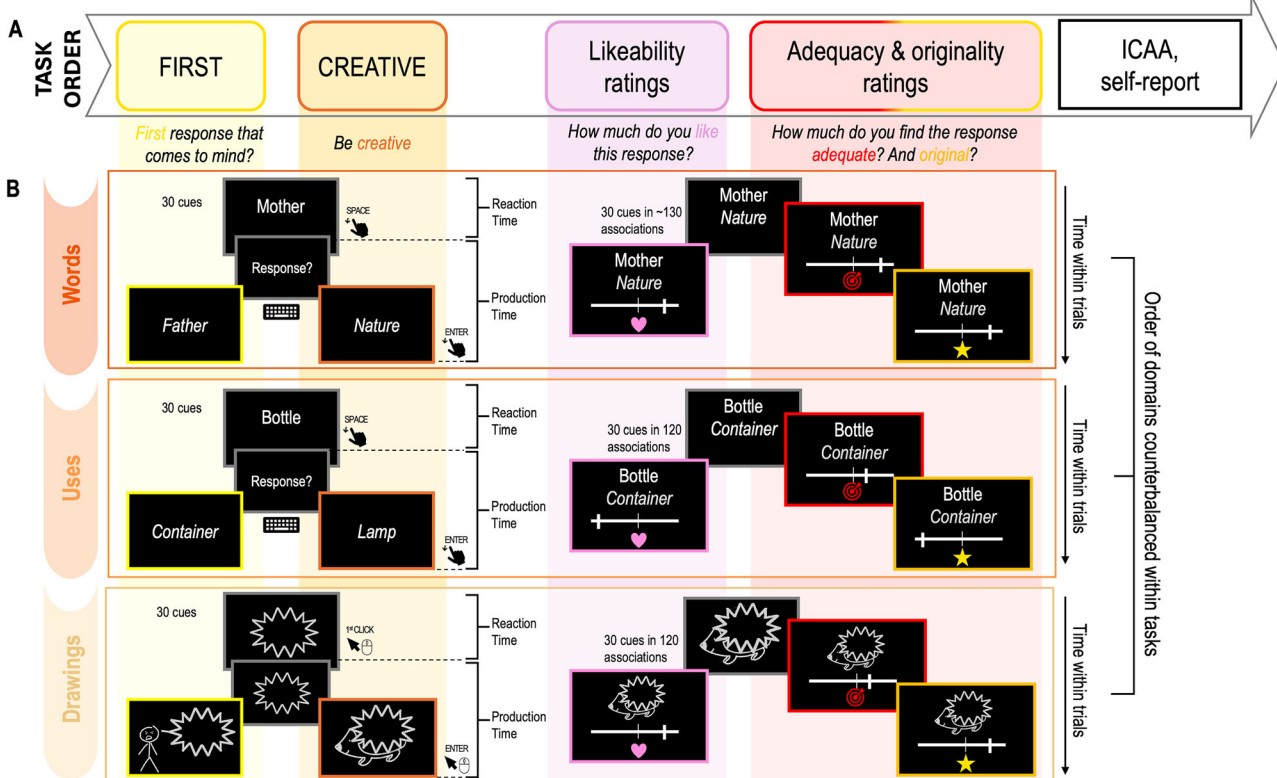

**Fig. 1 | Schematic representation of the experimental design. A** Task order. Participants performed the First condition of the Free Generation tasks. Then, they completed the Creative condition of these tasks. The generation tasks were then followed by the likeability rating tasks. Next, they completed the adequacy and originality rating tasks. Finally, they fulfilled a set of questionnaires (ICAA and self-report). **B** Details of the tasks for each domain: Words, Uses, and Drawings. Within each domain, the order of tasks was counterbalanced within and between participants. In the First condition (first column), participants were asked to provide the first response (word, use, or drawing) that came to mind when presented with a cue.

In the Creative condition (second column), they were instructed to provide creative responses to the same cues. Then (third column), participants saw their responses along with other potential responses and rated how much they liked each association (symbolized with a heart). Finally (fourth column), they rated the same associations for adequacy (symbolized with a target) and originality (symbolized with a star). The order of these ratings was randomized for each association. Participants' actions (keyboard pressing or mouse clicking) are symbolized with hands and mouse computer, and delimit timing variables (i.e., response and production times). Time that elapsed through the tests is symbolized vertically with arrows.

ratio[50], in which the variance of the additive parameters (α' and δ'), depending on the model, was set to zero such that they were considered fixed parameters. Priors were set to 0.5 for α parameters (no preference for originality or adequacy), to 1 for δ parameters (no convexity of preferences). The variance of priors was set to 10 for all free parameters. The procedure is defined as follows: 1) Multisource model fitting of the value function on the three domains' ratings: 3 sets of inputs (adequacy and originality within each domain) and 3 sets of outputs (likeability within each domain), with free parameters specific to each domain. 2) Definition of reduced models with all possible combinations for fixed or free parameters for each domain. 3) Bayesian Model Comparison based on the Savage-Dickey manipulation. Every possible combination of fixed and free additive parameters was tested, resulting in 13 models (see Supplementary Table 3). Note that implementing additional parameters (e.g., adding α' and δ') complexifies models and penalizes them for model selection.

**Canonical correlation.** Canonical correlation (or canonical variate analysis) is a statistical method to find the linear combination between two sets of variables that maximize the correlation inter-sets[51]. In other words, canonical correlation tries to maximize the association between the low-dimensional projection (i.e., canonical variable) of each set. To investigate the relation between model parameters and creativity questionnaires, we pooled variables in the Model & Behavior set (i.e., weighting parameter α and curvature parameter δ and creativity scores

from AI such as mean of SemDis, mean of AuDra, or mean of Word2Vec dissimilarity scores respectively for cue-responses associations in the Creative condition across the three domains) and Questionnaire set (i.e., ICAA scores and self-assessment of perceived level of creativity), and performed a canonical correlation between the two sets. Note that one participant did not fill the creativity questionnaires and therefore, was excluded from this analysis.

## Results

Seventy-three participants completed the First condition of three Free Generation tasks in a randomized order between domains (Words, Drawings, Uses), during which they provided the first responses that came to mind in reaction to a cue (word, object or abstract shape). Next, they completed the Creative condition of the generation tasks for the three domains, where participants were explicitly instructed to think creatively to produce a response associated with the cue, again in a randomized order. Then, they rated their responses regarding likeability, adequacy, and originality (see Fig. 1 and Methods). Finally, they completed creativity questionnaires (Inventory of Creative Activities and Achievements (ICAA), and self-report of perceived level of creativity).

First, we checked whether responses provided during the Creative conditions were indeed more creative than responses in the First conditions. Second, we tested whether the likeability of these responses correlated with response time and production speed within each domain. Third, we

conducted various model comparisons to determine whether participants' preferences similarly rely on adequacy and originality across different domains. Finally, we assessed whether the value parameters identified in the three domains were relevant to participants' creative abilities, as measured by the ICAA and self-report perceived level of creativity.

### Responses in the Creative conditions are more creative than in the First conditions

**Responses are both original and adequate in the Creative conditions.** Using adequacy and originality ratings provided by the participants in the three domains, we found that participants rated the responses from the Creative conditions as more original than those from the First condition (Words: $\text{Originality}_{\text{First}} = 35.10 \pm 1.70$ (Mean $\pm$ SEM), $\text{Originality}_{\text{Creative}} = 61.77 \pm 1.68$, First versus Creative: $t(72) = -14.18$, $p = 2 \times 10^{-22}$, $d = -1.84$, 95% difference CI = $[-30.42, -22.92]$; Uses: $\text{Originality}_{\text{First}} = 15.12 \pm 1.73$, $\text{Originality}_{\text{Creative}} = 68.31 \pm 1.36$, $t(72) = -26.66$, $p = 5 \times 10^{-39}$, $d = -3.98$, 95% difference CI = $[-57.17, -49.21]$; Drawings: $\text{Originality}_{\text{First}} = 46.20 \pm 1.86$, $\text{Originality}_{\text{Creative}} = 61.11 \pm 1.56$, $t(72) = -10.67$, $p = 2 \times 10^{-16}$, $d = -1.01$, 95% difference CI = $[-17.70, -12.13]$, one-sample two-tailed paired $t$ tests). Also, responses from the First condition were rated as more adequate than those from the Creative condition in the Words and Uses domains but no difference was observed in the Drawings domain (Words: $\text{Adequacy}_{\text{First}} = 82.79 \pm 1.28$, $\text{Adequacy}_{\text{Creative}} = 72.99 \pm 1.54$, First versus Creative: $t(72) = 7.33$, $p = 3.10^{-10}$, $d = 0.81$, 95% difference CI = $[7.13, 12.45]$; Uses: $\text{Adequacy}_{\text{First}} = 89.56 \pm 1.68$, $\text{Adequacy}_{\text{Creative}} = 64.55 \pm 2.00$, $t(72) = 9.86$, $p = 5 \times 10^{-15}$, $d = 1.58$, 95% difference CI = $[19.95, 30.07]$; Drawings: $\text{Adequacy}_{\text{First}} = 64.02 \pm 1.78$, $\text{Adequacy}_{\text{Creative}} = 64.91 \pm 1.51$, $t(72) = -0.63$, $p = 0.528$, $d = -0.06$, 95% difference CI = $[-3.68, 1.91]$). Importantly, the difference in originality ratings between First and Creative conditions was significantly higher than the difference in adequacy ratings for the three domains (for each task, one-sample two-tailed paired t-test between $\text{Adequacy}_{\text{First}}$-$\text{Adequacy}_{\text{Creative}}$ and $\text{Originality}_{\text{Creative}}$-$\text{Originality}_{\text{First}}$: Words: $t(72) = -9.41$, $p = 4 \times 10^{-14}$, $d = -1.20$, 95% difference CI = $[-20.46, -13.30]$; Uses: $t(72) = -10.51$, $p = 3 \times 10^{-16}$, $d = -1.43$, 95% difference CI = $[-33.52, -22.84]$; Drawings: $t(72) = -7.15$, $p = 6 \times 10^{-10}$, $d = -1.31$, 95% difference CI = $[-20.20, -11.40]$). Proportions of responses per bin of adequacy and originality ratings in each domain for First and Creative conditions are available in Supplementary Fig. 1, illustrating between-condition differences regarding adequacy and originality ratings. Altogether, these results indicate that for the participants, responses from the Creative condition were more original than those from the First condition (in the three domains) and that responses from the Creative condition were not significantly different in terms of adequacy in the Drawings domain, or slightly less adequate (in the Words and Uses domains) than the responses from the First condition. In other words, in the Creative condition, participants produced ideas they considered both adequate and original in the three domains.

**Creative responses take more time.** For all domains, response times in the Creative conditions were longer than in the First conditions (Words: $\text{RT}_{\text{First}} = 1.72 \pm 0.11$, $\text{RT}_{\text{Creative}} = 5.56 \pm 0.29$; Creative versus First: $t(72) = 14.05$, $p = 3 \times 10^{-22}$, $d = 2.02$, 95% difference CI = $[3.29, 4.38]$; Uses: $\text{RT}_{\text{First}} = 1.98 \pm 0.13$, $\text{RT}_{\text{Creative}} = 6.30 \pm 0.28$; $t(72) = 16.83$, $p = 1 \times 10^{-26}$, $d = 2.28$, 95% difference CI = $[3.81, 4.83]$; Drawings: $\text{RT}_{\text{First}} = 6.17 \pm 0.22$, $\text{RT}_{\text{Creative}} = 11.31 \pm 0.55$; $t(72) = 11.10$, $p = 3 \times 10^{-17}$, $d = 1.42$, 95% difference CI = $[4.21, 6.06]$). This result is consistent with the fact that producing responses with the aim to be creative takes more time[37,52].

**Objective scoring of creativity.** To address the creativity of responses in the three domains, we used pre-trained AI models. We estimated the semantic distance of responses to the cues using negative cosine similarity of a word2vec pre-trained model[53] in the Words domain, and using SemDis[43] in the Uses domain, as a proxy for creativity. For the Drawings domain, we used AuDra[42] to assign a creativity score to each drawing. The results indicate that

across all three domains, participants' responses were more creative (i.e., higher creativity scores for drawings, and higher dissimilarity between the cues and the responses for words and uses) in the Creative conditions compared to the First conditions (Words (word2vec dissimilarity): $\text{Prediction}_{\text{First}} = -0.326 \pm 0.007$, $\text{Prediction}_{\text{Creative}} = -0.163 \pm 0.007$; Creative versus First: $t(72) = 19.49$, $p = 2 \times 10^{-30}$, $d = 2.62$, 95% difference CI = $[0.15, 0.18]$; Uses (SemDis scores): $\text{Prediction}_{\text{First}} = 0.679 \pm 0.004$, $\text{Prediction}_{\text{Creative}} = 0.703 \pm 0.003$; $t(72) = 6.94$, $p = 1 \times 10^{-9}$, $d = 0.83$, 95% difference CI = $[0.02, 0.03]$; Drawings (AuDra scores): $\text{Prediction}_{\text{First}} = 0.484 \pm 0.002$, $\text{Prediction}_{\text{Creative}} = 0.501 \pm 0.003$; $t(72) = 10.07$, $p = 2 \times 10^{-15}$, $d = 0.82$, 95% difference CI = $[0.01, 0.02]$). Control analyses with alternative scoring method of Uses revealed similar results (see Supplementary Note 3) and AuDra scores were consistent with expert human ratings (Supplementary Note 4).

### Preferred ideas are provided faster across domains

To determine whether preferences energize the production of creative ideas consistently across all domains, we examined the relationship between the response times and the likeability ratings of responses from the Creative conditions. We analyzed two variables in the Creative conditions of the three domains: (1) the response time (RT), measured as the delay between cue display and the ready button press onset (i.e., space bar). Participants were instructed to indicate that they were ready to respond by pressing the space bar in the Words and Uses domains. In the Drawings domain, the first mouse click was used as the ready button press; (2) the production time, measured as the delay between the ready button press and the validation button press onset (i.e., return key). Participants were instructed to press the validation button when they were done responding. This production time allowed us to compute the production speed (PS), as the number of letters or pixels per second (see Methods). We hypothesized that those two-timing variables (i.e., RT and PS) should correlate with likeability ratings in Creative conditions as a signature of preferences in response production.

First, we regressed RT against likeability ratings for the Creative condition trials at the participant level. At the group level, we found significant negative relationships in each domain (Fig. 2A) (Words: $\beta = -0.11 \pm 0.03$, $t(72) = -4.45$, $p = 3 \times 10^{-5}$, $d = -0.52$, 95% CI = $[-0.16, -0.06]$; Uses: $\beta = -0.17 \pm 0.02$, $t(71) = -7 \times 10$, $p = 8 \times 10^{-10}$, $d = -0.82$, 95% CI = $[-0.22, -0.12]$; Drawings: $\beta = -0.25 \pm 0.03$, $t(72) = -9.03$, $p = 2 \times 10^{-13}$, $d = -1.05$, 95% CI = $[-0.30, -0.19]$, one-sample two-tailed $t$ test against zero).

Then, we regressed PS in the Creative conditions against likeability ratings in each domain (Fig. 2B). The relationship was significant and positive for the Words and Uses domains but did not reach significance for the Drawings domain (Words: $\beta = 0.13 \pm 0.03$, $t(72) = 4.68$, $p = 1 \times 10^{-5}$, $d = 0.54$, 95% CI = $[0.08, 0.19]$; Uses: $\beta = 0.24 \pm 0.03$, $t(71) = 8.51$, $p = 2 \times 10^{-12}$, $d = 0.99$, 95% CI = $[0.19, 0.30]$; Drawings: $\beta = -0.016 \pm 0.03$, $t(72) = -0.61$, $p = 0.542$, $d = -0.07$, 95% CI = $[-0.07, 0.04]$).

Strikingly, for response time, these results replicate the findings of Lopez-Persem et al.[21] in the Words domain and generalize them to object uses and drawings. This finding indicates that the relationship between preferences and response initiation in the Creative condition is consistent across domains. Nevertheless, the relationship between PS and likeability differed between the figural domain and the verbal domain. Post-hoc analyses revealed that the relationships between drawing time (DT) and drawing length (DL) (respectively the time between the first and last pixel drawn, and distance covered during drawing, the two variables to compute PS in the Drawings domain) with likeability ratings were significant (DT: $\beta = 0.20 \pm 0.03$, $t(72) = 6.64$, $p = 5 \times 10^{-9}$, $d = 0.77$, 95% CI = $[0.14, 0.26]$; DL: $\beta = 0.07 \pm 0.03$, $t(72) = 2.89$, $p = 0.005$, $d = 0.33$, 95% CI = $[0.02, 0.13]$, Fig. 2C). Thus, even though preferences do not seem related to the drawing speed as measured by PS, the time spent on the drawings and their length share some covariance with preferences.

Furthermore, we ran a series of control analyses to check whether confounding variables, including originality and adequacy, and proxy for confidence (i.e., quadratic representation of likeability) could influence the

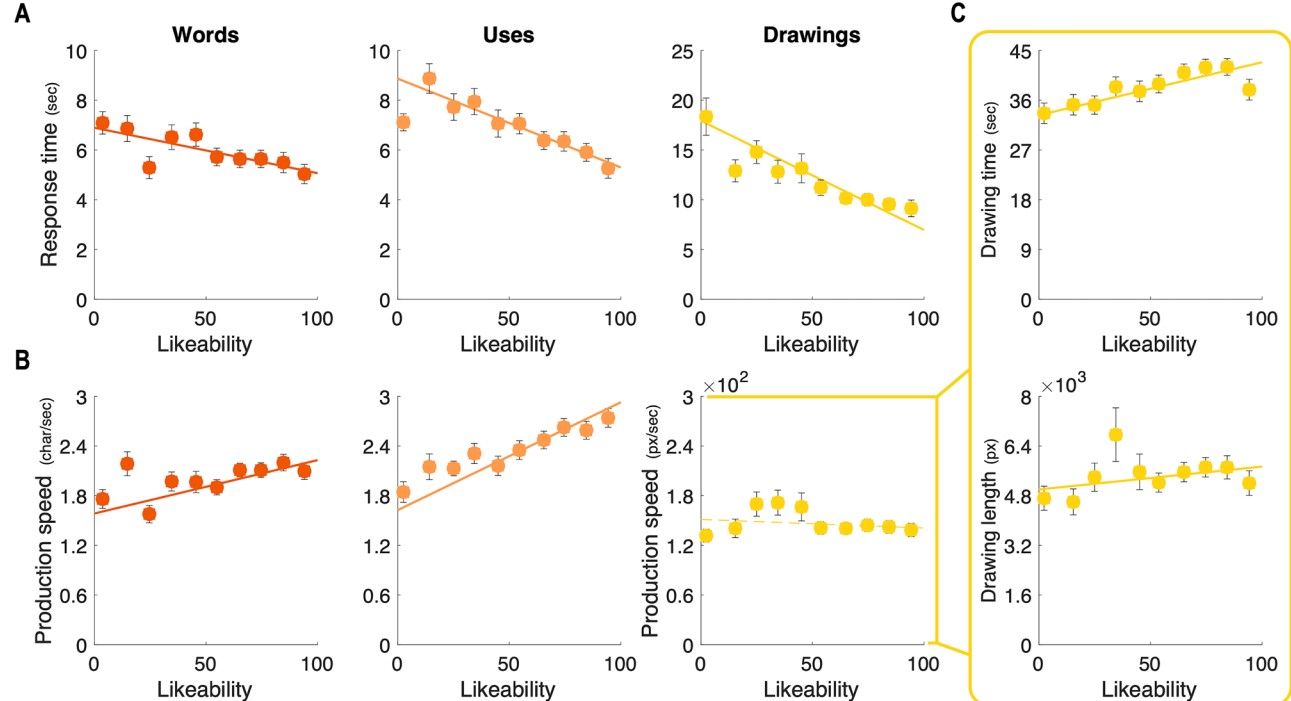

**Fig. 2 | Motivational effect: Relationship between likeability, RT, and PS across domains.** Relationships between response time (**A**) or production speed (**B**) and likeability ratings of responses from the Creative conditions in the Words (left, $n = 72$), Uses (middle, $n = 73$), and Drawings (right, $n = 73$) domains. Dots show binned data, averaged across participants. Data were grouped into ten bins based on likeability ratings, ranging from 0 to 100 in steps of 10. **C** Relationships between drawing time (top) or drawing length (bottom) and likeability ratings of responses from the Creative condition in the Drawings domain. Error bars represent inter-subject s.e.m. Solid lines correspond to the averaged linear regression fit of RT (**A**), PS (**B**), DT, and DL (**C**) against likeability ratings across participants, significant at the group level ($p < 0.05$). Dashed lines indicate non-significant regression coefficients.

relationship between the timing variables (i.e., RT and PS) and likeability ratings. The results remained robust, and these control analyses are available in Supplementary Tables 1 and 2 and Supplementary Notes 1 and 2.

## The likeability of responses relies on the same valuation process across the three domains of the generation tasks

For each domain, we observed that, on average, participants' ratings of likeability increased with both adequacy and originality (Fig. 3A). Based on this, we aimed to determine whether likeability judgments across the three domains relied on similar or distinct value functions. To do so, we used 12 value functions (see Methods) which combined adequacy and originality ratings to predict likeability judgments. We compared all possible combinations of value functions across the three domains, treating each combination as a model. These models were then grouped into two main families: the same value function explains likeability in all domains, or different value functions explain likeability across domains. Note that the second family includes models where each domain is explained by a different function, as well as models where two domains share the same function while the third uses a distinct one. The family model comparison revealed that the family of models using the same value function to explain likeability ratings across all domains provided a better fit compared to the family of models with different value functions (Same: Ef = 0.95, Xp = 1; Different: Ef = 0.048, Xp = 0 Fig. 3B).

Second, we found that the value function called Constant Elasticity of Substitution (CES)[54] provided the best fit for likeability ratings across the three domains, as indicated by the model comparison within the family for the same value function (Ef = 0.32; Xp = 0.72). This finding replicates previous results reported by Lopez-Persem et al. [21]. The CES value function (depicted in Fig. 3C) is characterized by two free parameters: the weighting parameter α, which captures the relative weight of originality (O) and adequacy (A) in preferences (i.e., how much one values originality over adequacy), and the convexity parameter δ, which captures the convexity of preferences, i.e., how much one values a balanced equilibrium

between adequacy and originality compared to an unbalanced equilibrium.

Finally, using the CES value function, we examined whether the same model's parameters could explain participants' likeability judgments across domains. To do so, we compared 13 versions of the CES function that account for preferences in the three domains (see Methods and Supplementary Table 3). The 13 versions vary in terms of number of free parameters. The simplest model has two free parameters (the same α and δ parameters across the three domains, i.e., the domain-general model), while the more complex model has 6 free parameters (α and δ specific for each domain, i.e., the domain-specific model). All possible combinations were tested as 11 alternative models (e.g., same α but different δ across domains). The group-level Bayesian Model Comparison showed that the domain-general model, with the same parameters α and δ across domains, provided the best fit (Fig. 3C; $Ef_{All=} = 0.74$; $Xp_{All=} = 1$). Details of the results for each model are reported in Supplementary Table 3.

We then investigated the values of the parameters $α_{general}$ and $δ_{general}$ estimated with the CES value function with equal parameter values across domains. On average, we found that the mean value of $α_{general}$ across participants was lower than 0.5, indicating that at the group level, participants tend to overweight adequacy compared to originality ($α_{general} = 0.44 \pm 0.02$ (Mean + SEM), $t(72) = -2.66$, $p = 1 \times 10^{-2}$, $d = -0.31$, 95% CI = [0.39,0.48], one-sample two-tailed $t$ test against 0.5). The parameter $δ_{general}$ was significantly lower than 1, which indicates a preference for compromise between adequacy and originality ($δ_{general} = 0.67 \pm 0.15$, $t(72) = -2.21$, $p = 0.03$, $d = -0.26$, 95% CI = [0.36,0.97], one-sample two-tailed $t$ test against 1).

To check the consistency of the estimated general parameters with those derived from each domain individually, we extracted parameter values from a CES fitting on each domain separately and performed an inter-individual correlation analysis. As expected, the general parameters strongly correlate with the parameters from each domain (Supplementary Fig. 2), as well as with the mean parameters across the three domains

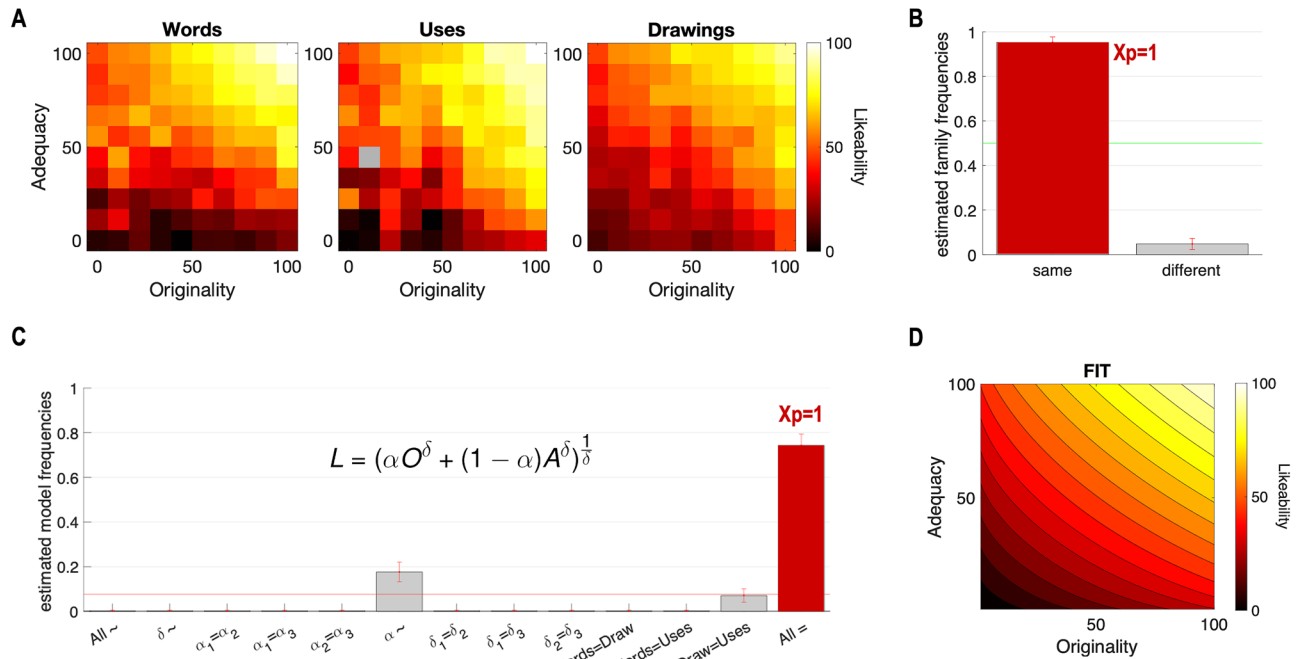

**Fig. 3 | Model generalization: Bayesian model comparison and selection of value functions across domains. A** Average likeability per bin of adequacy and originality ratings in each domain. The gray color indicates the absence of data for a bin. **B** Model comparison between Same and Different families of value functions. The bar plot compares the estimated frequency of the family of models where the same value function applies to all domains (same) versus the family of models where different value functions are used for each domain (different). **C** Comparison of CES value function (see equation) parameters' combination across domains. The bar

plots depict the estimated frequencies of models including different parameters (All ~), same parameters for the three domains (All =), or other possible combinations. ~ stands for the difference between the three domains, = stands for the same parameters for a pair of domains, index 1 stands for the Words domain, 2 for the Drawings domain, and 3 for the Uses domain. Xp stands for exceedance probabilities. Error bars depict posterior standard deviations. Red lines indicate the chance levels. **D** Heatmap representation of the data fit with the CES model All =. All participants were included in those analysis ($n = 73$).

$(r_{\alpha_{general},\alpha_{mean}}(70) = 0.88,\ p < 1 \times 10^{-6},\ 95\%\ \text{CI} = [0.82,0.93];$
$r_{\delta_{general},\delta_{mean}}(70) = 0.83,\ p < 1 \times 10^{-6},\ 95\%\ \text{CI} = [0.73,0.89];$ Supplementary Fig. 2). As a control analysis, we assessed the ability of the CES model to accurately recover parameter values using simulated likeability ratings (Supplementary Method 3 and Supplementary Fig. 3). This analysis revealed that the pre-defined and the estimated parameters of the simulations were highly correlated ($r_{\alpha_{pre-def},\alpha_{estimate}}(398) = 1,\ p < 1 \times 10^{-6},\ 95\%\ \text{CI} = [0.9994, 0.9996];\ r_{\delta_{pre-def},\delta_{estimate}}(398) = 0.93,\ p < 1 \times 10^{-6},\ 95\%\ \text{CI} = [0.92, 0.95]$), indicating a high degree of accuracy in parameter recovery.

### Model parameters and creative abilities

Finally, to investigate the relationships between the model parameters, the creativity of responses in the Free Generation tasks, and creative behavior measured by the questionnaires (ICAA and self-report of perceived level of creativity)[8], we used a canonical correlation approach. One set contained the $\alpha_{general}$ and $\delta_{general}$ model parameters and the mean creativity score per domain, based on the automatic scores (from the pre-trained models, see Methods) computed for each participant's response from the Creative conditions, as we want to investigate the relevance of the value parameters $\alpha$ and $\delta$ to creativity. This first set, named Model & Behavior, was compared to the second set composed of ICAA scores and self-report of creativity level (Questionnaire set, see Methods). The first canonical variable raised a significant correlation ($r(70) = 0.55$, Wilks statistic = 0.57, $F(15,177.1) = 2.62$, $p = 0.001$), highlighting a shared variance between the two sets. Interestingly, all variables of the Questionnaire set contributed significantly to its associated canonical variable (Questionnaire set: $r_{\text{C-Ach}}(70) = 0.78$, $p = 1 \times 10^{-15}$, 95% CI = [0.66,0.85]; $r_{\text{C-Act}}(70) = 0.76$, $p = 9 \times 10^{-15}$, 95% CI = [0.64,0.84]; $r_{\text{C-self-report}}(70) = 0.90$, $p = 8 \times 10^{-27}$, 95% CI = [0.84,0.94] Pearson correlation coefficient between variables and canonical variable of the set) (Fig. 4). In the Model & Behavior set, model's parameters

($r_{\alpha}(70) = 0.43$, $p = 1 \times 10^{-4}$, 95% CI = [0.22,0.60]; $r_{\delta}(70) = -0.38$, $p = 9 \times 10^{-4}$, 95% CI = [−0.57,−0.17]) contributed significantly to their associated canonical variable. Regarding automatic scoring of creativity, both the AuDra score ($r_{\text{Drawings - AuDra}}(70) = 0.72$, $p = 1 \times 10^{-12}$, 95% CI = [0.58,0.81]) and the Word2Vec score (negative cosine similarity) ($r_{\text{Words - Word2Vec}}(70) = 0.46$, $p = 5 \times 10^{-5}$, 95% CI = [0.25,0.62]) contributed significantly and positively to the associated canonical variable. The SemDis score did not show any significant contribution ($r_{\text{Uses - SemDis}}(70) = -0.09$, $p = 0.43$, 95% CI = [−0.32,0.14]). Importantly, we can note that both value parameters contribute to their canonical variable, with a positive relationship for $\alpha$ and a negative relationship for $\delta$: individuals favoring originality and compromises for an equilibrium of originality and adequacy score higher on ICAA and creativity self-report. Control analyses using an alternative scoring method of participants' responses in the Uses domain are detailed in Supplementary Note 3 and Supplementary Fig. 4.

### Discussion

In the current study, we explored subjective valuation in creative idea production across three domains of creativity: semantic association (Words domain), figural construction (Drawings domain), and concept association/utilitarian creativity (Uses domain). Our findings first showed that valuation has a motivational role in creativity, whether it serves to energize simple actions (such as button presses) or induce the allocation of resources for more complex endeavors (such as drawings). Second, we found that the valuation of ideas relies on a combination of their adequacy and originality and that the individual parameters that govern this combination relate to creative behavior and self-perception of creativity in real life. Finally, and crucially, this study revealed that the same value function explains the valuation of ideas across three domains of creativity, with similar value parameters across domains. Overall, these findings demonstrated that creativity involves a domain-general valuation process.

**Fig. 4 | Relevance of model parameters and behavioral variables regarding creative abilities.**
**A** Correlation between the first canonical variable of the Questionnaire set and the first canonical variable of the Model & Behavior set. **B** Bar plots represent the correlation coefficient between each variable of the Model & Behavior set with the first canonical variable from the Model & Behavior set. For Words, Uses, and Drawings, variables are individual means of respective scores. **C** Same as (**B**) for each variable of the Questionnaire set with the first canonical variable from the Questionnaire set. Stars indicate the significance of the correlation ($p < 0.05$). One participant was excluded of this analysis due to a lack of data ($n = 72$).

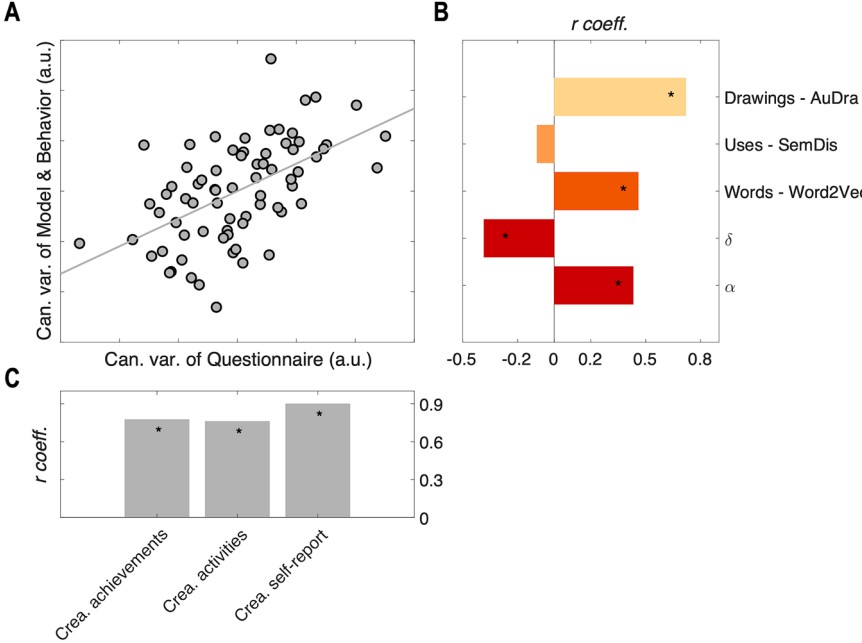

We found a negative relationship between response time and the likeability of responses across all domains of creativity, suggesting a motivational effect: the more participants like an idea, the faster they initiate it. This replicates previous findings[21,25] and generalizes them to figural and utilitarian creativity. Additionally, likeability positively correlates with production speed in verbal tasks (i.e., semantic and concept/uses associations) consistent with Lopez-Persem et al.[21], and generalizing it to the present inspired version of the AUT. However, in the Drawings domain, we found that likeability positively correlates with both drawing time and length, indicating that preferred drawings tend to be longer and require more time to produce. These results suggest that participants invest more effort and time into producing the drawings they prefer, consistent with the motivation interpretation. In summary, energization in verbal tasks and effort in drawing production can be seen as two sides of the same coin, both reflecting resource allocation. In some contexts, resource allocation is associated with an increased production speed (e.g., typing responses on a keyboard) or duration (e.g., drawing with a computer mouse). Resource allocation is especially relevant regarding the effortful process that is creativity[52], where achieving higher originality requires more effort[37]. This mirrors findings from the effort allocation literature, where individuals exert more effort for greater rewards[26,55]. Finally, these differential motivational effects in the figural domain highlight the need for further research, as motivation may drive different forms of effort or engagement depending on the creative domain.

Altogether, these markers of preferences during creative production reinforce the role of valuation in creativity. They provide deeper insight into the common, domain-general mechanism underlying the well-established relationship between motivation and creativity[22], which appears to play a substantial role across domains. Indeed, the Componential theory of creativity[56,57] highlights the performative role of both extrinsic motivation (e.g., social environment) and intrinsic motivation (e.g., affect, sense of challenge) in creativity. Furthermore, results from one of our recent studies have shown that individuals who reported feeling more motivated during the first COVID-19 lockdown (compared to before) also perceived themselves as more creative across domains[58].

Using Bayesian model fitting and comparisons, we found that likeability judgments rely on a combination of adequacy and originality judgments. In particular, we identified the Constant Elasticity of the Substitution (CES) value function[54] as the best model to explain likeability ratings, replicating previous results[21]. This function combines adequacy and originality with a relative weighting parameter (α) and allows non-linear

interaction through a convexity parameter (δ). Crucially, our results demonstrate that likeability judgments are better explained by a CES function with common α and δ across the three domains than by a model with different parameters for each domain. This key finding supports the central hypothesis of this study regarding the domain generality of the valuation process of creativity.

Our results align with research on preference stability in non-creative contexts showing that preferences tend to remain stable across tasks when decisions involve stimuli with similar attributes. For example, one study demonstrated that the incorporation of costs and rewards into decision remains stable across different types of costs[27]. Another study found that attributes for altruistic decisions are integrated similarly across different measurement methods, such as ratings, choices, or physical effort allocation[26]. Second, attributes are often specific to domains. For instance, one study found that distinct attributes are used when making choices in different domains (i.e., the main attributes for movies and food are different)[59]. Despite these results, preference stability across domains remains largely unexplored. In our study, one could have expected that aesthetic considerations in the Drawings domain might have reduced the contribution of adequacy and originality. However, our results revealed that the same dimensions (adequacy and originality) continue to account for preferences equally across all domains examined. This finding highlights the pivotal role of originality and adequacy in the evaluation process of creativity and suggests that these two attributes are among the most important for assessing any type of idea in a creative context.

The canonical correlation analysis revealed that both α and δ parameters share variance with creative behavior and self-perceived creativity levels. This result demonstrates that domain-general value parameters can partially predict creative abilities, aligning with the results of Lopez-Persem et al.[21] and Moreno-Rodriguez et al.[25] in the semantic association domain and generalized them to two additional creativity domains, including figural creativity. Hence, the parameters of our model, estimated at the crossroads of three creative domains, appear to be ecologically relevant to creative behavior.

While the current results offer evidence that idea evaluation involves a domain-general valuation process, the domain-generality or specificity of the idea generation phase remains unaddressed. The generation phase of creativity has been mostly studied in verbal creativity, where it has been linked to the organization of semantic associations in memory[60,61]. According to the associative theory of creative ideation, idea generation involves a combination of knowledge and concepts stored in semantic memory[62]. A recent study, by

He et al.[63] suggests that the generation phase is domain-specific. Their findings indicate that verbal creativity, rather than figural creativity, is correlated with associative abilities, highlighting the importance of semantic memory structure in verbal creativity but not in figural creativity. Overall, it is likely that idea generation is influenced by knowledge and abilities specific to the domain of the problem, supporting domain-specificity. The previously developed Generation-Monitoring-Valuation-Selection (GMVS) model by Lopez-Persem et al.[21] conceptualized idea generation as a generator module which explores semantic space using a frequency-driven random walk within an internal repertoire (a network of words connected by semantic proximity —i.e., frequency of association). However, the internal repertoire of concepts and knowledge could be modeled not only as a network of interconnected words but also as a network of any type of representation, including images[64,65]. Within the GMVS model framework, we might expect that the explored internal repertoire would be dominated by semantic characteristics in the Words task and by figural characteristics for the Drawings task. In the AUT, a more mixed repertoire might be explored, incorporating both semantic and figural aspects of the cues. Future studies should investigate the domain-generality or domain-specificity of the generation phase, by comparing models that use domain-specific concept representations versus those integrating multi-domain concept representations[65]. In addition, it will be necessary to compare exploration rules such as biased random walks[66] or trajectories following the optimal foraging theory[67] across different domains.

A recent framework, consistent with the two-stage view of creativity, proposes that episodic and semantic memory contributes to the generation but also the evaluation stage[68]. This implies that domain-specific knowledge can influence both the generation and evaluation phases of creativity. Our findings are not inconsistent with this view. Here, we do not claim that judgments of originality and adequacy are inherently domain-general[69–71]. Rather, we argue that the process by which originality and adequacy are integrated—regardless of their accuracy—is domain-general. In other words, our results support the hypothesis that the mechanism governing this integration is consistent across domains. However, these results do not exclude the potential influence of memory and past experiences in a specific domain on other processes of the evaluation stage. Domain expertise, which is closely linked to memory and past experience, may influence the way adequacy and originality are monitored and assessed. For instance, experts might possess a richer set of reference points against which they can assess the originality and adequacy of ideas within their field. Consequently, experts might more accurately assess adequacy and originality, potentially enhancing their creative performance within their field. Recent experimental findings indicate that experts outperform novices in creative tasks directly related to their field of expertise (e.g., music improvisation[72]), yet, no significant differences between novices and experts were observed on classic divergent thinking tasks, such as the AUT[11,33]. Thus, expertise and domain knowledge are better predictors of domain-specific creativity[73], but it is unclear whether this advantage arises from better idea generation, better monitoring, or both. The present study did not formally address the relationship between domain expertise and creativity. A promising direction for future studies would be to test the domain-generality of valuation in creative experts across different domains to determine whether expertise influences how creative ideas are liked and how their originality and adequacy are assessed.

At the neural level, future neuroimaging studies could investigate the neural representation of generation and valuation processes across different creative domains. A recent fMRI study[25] using the GMVS framework demonstrated the involvement of the Brain Valuation System (BVS), which encodes the likeability of ideas during their production. Based on our results and in the light of this literature, we propose two hypotheses: (i) the BVS will represent the likeability of ideas across all domains, given its generic property[74], i.e., it represents likeability of any kind of item; and (ii) the exploration of the internal repertoire may differ between creative domains, depending on whether the creative output is semantic or figural. This domain-dependent exploration might be reflected in distinct brain mechanisms, supported by distinct brain regions[15,20] or different activity patterns within a given network.

## Limitations

Several limitations can be highlighted in our study. First, the concept of likeability is open to discussion. Participants rated how much they liked their responses in the context of the Creative conditions, with the likeability reflecting both the goal value (being original and adequate) and the hedonic value of the ideas, which may depend on other attributes, such as, for instance, drawings' aesthetics[75,76]. Our experimental design does not allow for a clear dissociation between hedonic and goal value components of responses, as it has been achieved in previous studies in the field of value-based decision-making[77]. If the goal value is what is captured by our value function, it raises the question of the role played by the hedonic component. To capture this component, we orthogonalized likeability ratings by adequacy and originality (equivalent to removing the goal-related components from likeability). Interestingly, the correlation between likeability and response time in the Creative conditions remained, suggesting that the hedonic value component of likeability ratings may also play a role in creative idea production. A second limitation is that likeability ratings might also involve metacognitive components[78]. Although we controlled for potential confounds between likeability and confidence in control analyses, by using squared likeability as a proxy for confidence, following prior methods[79], the distinction between likeability and confidence in responses remains to be clarified. Future studies are needed to better understand the factors contributing to likeability ratings. Finally, another limitation of the present protocol concerns the lack of a significant contribution of the SemDis score associated with Uses responses to the canonical variable related to the task (i.e. Model & Behavior set). The AUT typically involves producing several alternate uses of the same object within several minutes, depending on the version[32]. In our task inspired by the AUT, however, the maximum time allowed in the Creative condition was limited to 20 s, and participants were instructed to provide only a single response. These modifications may have reduced the opportunity for participants to produce responses with sufficient variability in creativity, potentially affecting the observed contribution of the SemDis scores. Note that alternative automatic scoring methods such as OCSAI[44] yielded similar non-significant results (see Supplementary Note 3).

In this study, we demonstrated that individual preferences during creative ideation can be modeled by the same value function with the same parameters across creative domains, including semantic association, figural construction, and object uses. This finding indicates that creative idea valuation is a domain-general mechanism, with stable creativity-related preferences across domains. Moreover, all creative domains appear to be influenced by motivation, suggesting that the valuation of an idea modulates resource allocation during creative behavior. Overall, our findings provide some of the first experimental and computational evidence supporting a domain-general subprocess of creative thinking.

## Data availability

The authors declare that all collected and preprocessed data are available in the following OSF repository: https://osf.io/pykvf/[80].

## Code availability

All codes are resources supporting the findings of this study are available in the following OSF repository: https://osf.io/pykvf/[80].

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

## Acknowledgements

E.V. is funded by the Agence Nationale de la Recherche [grant numbers ANR-19-CE37-00153 01]. The research also received funding from the program Investissements d'avenir ANR 10- IAIHU-06. A.L.P. was supported by the Fondation des Treilles. The study was funded by the European Union's Horizon 2020 research and innovation program under the Marie Sklodowska-Curie grant agreement No 101026191. G.B. was financed by the Ecole Doctoral Cerveau, Cognition, Comportement (ED158) affiliated with Sorbonne University. The funders had no role in study design, data collection and analysis, decision to publish or preparation of the manuscript. We thank all the participants in the study, and the PRISME platform for helping in data collection. Also, specific gratitude is directed toward the members of the CreaTeam for their support in this project, especially for their ratings of participants' creative outputs.

## Author contributions

E. Volle, A. Lopez-Persem, and G. Battistello designed the study. G. Battistello collected the data and performed all analyses. S. Moreno-Rodriguez and A. Lopez-Persem provided template scripts for analyses. A. Lopez-Persem and G. Battistello conceptualized the computational models. G. Battistello, A. Lopez-Persem, and E. Volle wrote the article. All authors reviewed and edited the article.

## Competing interests

The authors declare no competing interests.
