## [Transparent Peer Review file · Communications Psychology]

Subjective valuation as a domain-general process in creative thinking

Corresponding Author: Mr Gino Battistello

Version 0:

Decision Letter:

Dear Mr Battistello,

Thank you for your patience during the peer-review process, and my sincere apologies for the delay. Your manuscript titled "Subjective valuation as a domain-general process in creative thinking" has now been seen by 3 reviewers, and I include their comments at the end of this message. They find your work of interest but raised some important points. We are interested in the possibility of publishing your study in Communications Psychology, but would like to consider your responses to these concerns and assess a revised manuscript before we make a final decision on publication.

We therefore invite you to revise and resubmit your manuscript, along with a point-by-point response to the reviewers. Please highlight all changes in the manuscript text file.

Editorially, we consider it important that the concerns regarding the reliability of the scoring technique are thoroughly addressed (see comments by Reviewer 1 and 2 for further guidance).

I am attaching an Editorial Requests Table that details critical reporting requirements for the revised manuscript. Please attend to each item and ensure your manuscript is fully compliant. We are requesting that your manuscript aligns with these requirements as this facilitates the evaluation of your manuscript, reducing delays in re-review and potential future acceptance. If your revised manuscript is not aligned with these requests on major issues, such as those concerning statistics, it may be returned to you for further revisions without re-review. Additional information can be found in our style and formatting guide <https://www.nature.com/documents/commspsychol-style-formatting-guide-accept.pdf> Communications Psychology formatting guide.

Please use the following link to submit your

- revised manuscript,
- point-by-point response to the referees' comments,
- cover letter (as a separate document),
- the Editorial Policy Checklist (see below),
- the Reporting Summary (see below), and
- the completed Editorial Request Table (attached):

Link Redacted

We hope to receive your revised paper within 8 weeks; please let us know if you aren't able to submit it within this time so that we can discuss how best to proceed. If we don't hear from you, and the revision process takes significantly longer, we may close your file. In this event, we will still be happy to reconsider your paper at a later date, provided it still presents a

significant contribution to the literature at that stage.

Best regards,

Nicole Montijn

Dr Nicole Montijn
Consulting Editor
Communications Psychology

REVIEWER EXPERTISE:

Reviewer #1: Creativity; Fluid Intelligence; Computational Modeling

Reviewer #2: Creativity; Mind Wandering; Computational Modeling

Reviewer #3: Creativity; Emotion; Cognitive Neuroscience

REVIEWER REPORTS:

Reviewer #1 (Remarks to the Author):

This is an interesting and important study showing that idea evaluation is a domain-general cognitive ability. I found the paper to be well-written and accessible, which is notable given the computational modeling. Still, I have a few points I would like the authors to consider, particularly around the method/creativity scoring approaches (and one that I see as potentially problematic). I see the points as addressable and hope to see this work published.

Some of the language could be updated, e.g., abstract "It is admitted that creativity..."

Also "This is consistent with study14" which I think has to do with the numbered referencing style.

The introduction provides good coverage of relevant literature. However, one thing that is missing is a discussion of the role of expertise, which is particularly relevant for creativity in specific domains, especially Big C creativity, albeit this study focused on lab tasks that do not require expertise (knowledge and skills).

Results: the authors used SemDis to analyze the AUT. However, the correlation of semdis to human ratings is not as strong as more recent approaches, such as LLMs trained to predict human ratings - did the authors consider using this approach as it is more robust? Even though responses are in French, they can be scored by LLMs (<https://openscoring.du.edu/scoringllm>). This may explain relatively lower correlations between model parameters for alpha (cf. supplemental fig 2). Also, I may have missed it but I didn't catch zero-order correlations between originality ratings between the three tasks?

Organisciak, P., Acar, S., Dumas, D., & Berthiaume, K. (2023). Beyond semantic distance: Automated scoring of divergent thinking greatly improves with large language models. *Thinking Skills and Creativity*, 49, 101356.

Figure 1 was a lot to process. I appreciate the effort but I wonder if this could be made more simple/easier to digest?

Figure 2 is labeled "Motivation Effect" but the text does not really refer to it as such. In fact, the term "motivation" doesn't appear until the Discussion, so I would discuss it sooner (though I know "energizing" has been used earlier). This could make things more clear.

Figure 4, I suggest making panel C y axis more granular than just 0, .5, 1.

I think the Discussion would also benefit from mentioning the potential role of expertise, as noted earlier, beyond lab tasks to real-world tasks requiring expert knowledge and skills.

Can the authors explain why they used drawing stimuli that were not in the original training set of AuDrA? The paper shows that the model does not perform well on drawing stimuli that it was not trained on. I would also like to see some validation evidence to show the drawing scores worked as expected; I suspect the values may be driven by pixel count/elaboration. Also, why could participants not erase what they drew?

<https://link.springer.com/article/10.3758/s13428-023-02258-3>

Reviewer #2 (Remarks to the Author):

The present manuscript, 'Subjective valuation as a domain-general process in creative thinking' addresses the question of whether or not creative valuation is a domain general process. Findings suggest a consistent valuation mechanism across word association, object uses and drawing tasks. Authors make the case that creative evaluation is inherently domain general. While the topic is of general interest and the study is well presented, more work is needed to provide background on previous research in this area and soften the claims of domain generality. Additionally, there are several instances throughout the manuscript where language could be improved for clarity and precision. See below for specific comments:

- On p. 3, line 51, authors cite findings from Reiter-Palmon and colleagues (2009) to support that divergent thinking tasks have predictive validity of real-world creative achievement. What about studies looking at DT performance in creative experts? There is more work on this topic which should be referenced here, expanding on the domain-generality of divergent thinking in real-world creative people.

- Moreover, in line 73, authors outline the dual-process model of creativity. A recent paper by Benedek et al. (2023) provides a detailed model of this framework and the contributions of memory to the generative/evaluative stages of creativity. The introduction of the present manuscript would be improved by a brief discussion of this framework, as a model for the creative process.

- The schematic representation in Figure 1 is overly complex and difficult to interpret. I recommend simplifying this figure to be more readily interpretable.

- In the results, authors report that participants rated responses from the creative condition as being more original than the first responses. Could it be the case that there is a bias in how participants rate their responses in the two different conditions. In other words, that the responses do not vary in their originality, but there is some bias in how participants are ratings their own responses across these conditions?

- On p. 8, line 179, authors state, "this result is consistent with the fact that producing creative responses takes more time.' A citation is needed here to support this claim.

- On line 184 authors describe the use of semantic distance as a proxy for creativity assessment. While these approaches are informative, the Open Creativity Scoring platform significantly outperforms these word2vec models and is generally accepted as a more reliable method for scoring originality of AUT responses. Is there a reason why the authors opted for semantic distance approaches over this Open Scoring method? If not, I strongly recommend repeating the analysis with this approach (see reference below).

Organisciak, P., Dumas, D., Acar, S., and de Chantal, P. L. (2024). Open Creativity Scoring [Computer software]. Denver, CO: University of Denver. <https://openscoring.du.edu>.

Reviewer #3 (Remarks to the Author):

This manuscript describes a study investigating whether the evaluation process in creativity is domain-general or specific. They asked 73 participants to complete three different creativity tasks and evaluate their output, across three domains: verbal creativity, drawing and utilitarian or conceptual creativity. They also asked them to complete the ICAA – an inventory of creative activities and achievements. The authors hypothesized that (i) creative idea production is energized by valuation across domains, (ii) the construction of subjective values is based on both the adequacy and originality of response, also similarly across domains, (iii) preferences are stable across domains and (iv) correlated with creative abilities in real-life.

They found that valuation increased motivation across domains, in line with their first hypothesis. They also found that preferences are stable across domains and correlated with creative abilities in real-life. Finally, the authors found that valuation is indeed based on both originality and adequacy, regardless of domain.

In both semantic and utilitarian creativity, positive valuation was positively correlated with speed of responses – that is, the more one judged an item as valuable, the faster the response (or the shorter the response time). For drawing, however, preferred drawings tended to take longer to produce. This is an interesting distinction that merits further study, as effort and motivation might result in differential engagement in specific domains. Future directions should include neuroimaging as a way of better understanding what underlies these differences.

Overall, this paper is thorough and important, adequately powered, and carefully analyzed. More work is needed, however, to improve the clarity and general understandability of the arguments and results. I have made some notes below, but the authors should carefully re-read their manuscript to align tenses, and improve wording.

With these improvements, I believe this manuscript would make a positive addition to the literature and is worthy of publication.

Page 1, line 14: 'It is admitted' change to Experts agree that

Line 16: "The evaluation phase involves valuation processes upstream selection" not sure what this means

Line 17: valuation of ideas not value?
Line 55: Perspectives in the field - is that a specific reference? Why is it past-tense?
Line 61: achievements 'tend' (remove 's')
Line 63: with a study...
Line 70: Remove 'a' before 'generation' and 'evaluation'
Line 78: align tenses
Line 79: 'The authors found' rather than 'it was'
Throughout that paragraph, align tenses.
Line 116: 'Alternate uses, not alternative', throughout the manuscript
Line 415: 'significantly better explains' is awkward
Line 417: add 'literature' or 'research' to 'past'
Line 465: remove 'intuitively'

EDITORIAL POLICIES

We ask that you ensure your manuscript complies with our editorial policies and reporting requirements.

To that end, we require revised manuscripts to be accompanied by two completed items: a reporting summary that collects information on study design and procedure, and an editorial policy checklist that verifies compliance with all required editorial policies.

- <https://www.nature.com/documents/nr-reporting-summary.zip>>Nature Research Reporting Summary
- <https://www.nature.com/documents/nr-editorial-policy-checklist.pdf>>Editorial Policy Checklist

All points on the policy checklist must be addressed. Your revised manuscript can only be sent back to the referees if these checklists are completed and uploaded with the revision.

Notes: If you have submitted a Stage 1 Registered Report, Review, Primer, Comment, or Perspective you do not need to submit these forms. If you have already submitted these forms, you may disregard this request.

Communications Psychology is committed to improving transparency in authorship. As part of our efforts in this direction, we are now requesting that all authors identified as 'corresponding author' create and link their Open Researcher and Contributor Identifier (ORCID) with their account on the Manuscript Tracking System prior to acceptance. ORCID helps the scientific community achieve unambiguous attribution of all scholarly contributions. You can create and link your ORCID from the home page of the Manuscript Tracking System by clicking on 'Modify my Springer Nature account' and following the instructions in the link below. Please also inform all co-authors that they can add their ORCIDs to their accounts and that they must do so prior to acceptance.
<https://www.springernature.com/gp/researchers/orcid/orcid-for-nature-research>

Version 1:

Decision Letter:

Dear Mr Battistello,

Your manuscript titled "Subjective valuation as a domain-general process in creative thinking" has now been seen by our reviewers, whose comments appear below. In light of their advice I am delighted to say that we are happy, in principle, to publish a suitably revised version in Communications Psychology.

We therefore invite you to revise your paper one last time to address the remaining concerns of our reviewers and a list of editorial requests. At the same time we ask that you edit your manuscript to comply with our format requirements and to maximise the accessibility and therefore the impact of your work.

EDITORIAL REQUESTS:

SUBMISSION INFORMATION:

OPEN ACCESS:

*** TRANSPARENT PEER REVIEW:** Communications Psychology uses a transparent peer review system. On author request, confidential information and data can be removed from the published reviewer reports and rebuttal letters prior to publication. If you are concerned about the release of confidential data, please let us know specifically what information you would like to have removed. Please note that we cannot incorporate redactions for any other reasons.

*** CODE AVAILABILITY:** All Communications Psychology manuscripts must include a section titled "Code Availability" at the end of the methods section. We require that the custom analysis code supporting your conclusions is made available in a publicly accessible repository at this stage; please choose a repository that generates a digital object identifier (DOI) for the code; the link to the repository and the DOI must be included in the Code Availability statement. Publication as Supplementary Information will not suffice.

*** DATA AVAILABILITY:**

Link Redacted

Best regards,

Marieke, on behalf of

Nicole Montijn

Nicole Montijn
Editor
Communications Psychology

Marike Schiffer, PhD
Chief Editor
Communications Psychology

REVIEWERS' COMMENTS:

Reviewer #1 (Remarks to the Author):

The authors were very responsive to my previous comments, which I appreciate. I have nothing further to add and congratulate them on an excellent piece of work.

Reviewer #2 (Remarks to the Author):

Thank you for comprehensively addressing all points raised. I have no additional comments.

Reviewer #3 (Remarks to the Author):

The authors have adequately addressed the concerns that I and the other reviewers raised and I'm pleased to say that I feel the manuscript is now ready for publication.

Response to referees

We thank the reviewers for their positive and constructive assessments of the manuscript, leading to its consequent improvement. We reply to comments and suggestions on a point-by-point basis. Please note that changes made to the manuscript are highlighted in yellow when related to language improvement and green when related to the reviewer's comments.

The modifications made to the manuscript can be summarized as follows:

- Language improvement
- Addition of control analyses regarding creativity scoring of Uses using OSCAI instead of SemDis
- Validation of AuDra scores with the collection of human ratings of the creativity of drawings
- Discussion of the role of expertise and memory
- Clarification of the terms related to motivation
- Reformatting of the task figure

Color-code

Black: Reviewers' comments

Blue: Authors responses

Green: Authors citation from the manuscript (unedited)

Green highlighting on green font: Edits in the manuscript related to reviewers' comments.

Yellow highlight on green font: Edits in the manuscript related to grammar and language improvements.

Reviewer #1

This is an interesting and important study showing that idea evaluation is a domain-general cognitive ability. I found the paper to be well-written and accessible, which is notable given the computational modeling. Still, I have a few points I would like the authors to consider, particularly around the method/creativity scoring approaches (and one that I see as potentially problematic). I see the points as addressable and hope to see this work published.

We thank the reviewer for their overall assessment of our manuscript. We have carefully addressed their comments.

1. Some of the language could be updated, e.g., abstract "It is admitted that creativity..." Also "This is consistent with study14" which I think has to do with the numbered referencing style.

We thank the reviewer for this comment. We have updated the manuscript after checking for language clarity and referencing, including the abstract. All revisions have been highlighted in yellow in the manuscript. For instance, the abstract has been updated as:

(Abstract – L12 to 19)

Is a talented painter also a proficient writer? The ongoing discourse on whether creativity operates through domain-general or domain-specific mechanisms has led to challenges in our understanding of the creative process. Prior

research suggests that creativity comprises two phases: idea generation and evaluation. A recent framework has proposed that the evaluation phase involves a valuation process which occurs upstream of the selection of an idea. In this framework, the value assigned to an idea, i.e., how much one likes an idea, energizes its production and drives its selection. While the role of valuation has been demonstrated in verbal semantic creativity, its domain generality remains to be tested. In this study, we assessed whether valuation is a domain-general or domain-specific process. Seventy-three participants engaged in three creativity tasks (producing semantic associations, alternate object uses, and drawings) followed by rating tasks. Using computational modeling, we found that a consistent valuation mechanism governs idea valuation across different domains. Specifically, the same value function and value parameters were shared across the evaluation of word associations, object uses and drawing completions. These findings advance our understanding of the evaluation phase of creativity, portraying the valuation component as inherently domain-general. Identifying such core components of creative ideation contributes to elucidating the cognitive mechanisms underlying creativity and provides empirical support for including valuation as a core process in creativity.

2. The introduction provides good coverage of relevant literature. However, one thing that is missing is a discussion of the role of expertise, which is particularly relevant for creativity in specific domains, especially Big C creativity, albeit this study focused on lab tasks that do not require expertise (knowledge and skills).

We thank the reviewer for their comment. Our study focuses on little-c creativity, but the role of expertise could be indeed relevant in our context. We have addressed this point in the introduction of the manuscript (as this point was also raised by Reviewer 2 in their Comment #1), by including the following paragraph:

Introduction (L56 to 65)

For instance, scores of creative achievements and activities in the literature domain tend to correlate with both achievements and activities in a wide range of domains, such as culinary and visual arts, but the highest correlation remains for creative writing itself. These findings from questionnaires are consistent with empirical evidence⁹ indicating that life experiences, such as education and cultural background, have a stronger impact on domain-specific creativity than on domain-general creativity. Overall, these findings suggest that creativity has both domain-general and domain-specific mechanisms, the latter potentially developed from previous experiences and domain expertise^{10,11}. A key remaining question is the extent to which the core cognitive mechanisms underlying creative thinking are domain-general or domain-specific.

Additionally, we have checked within our data whether ICAA domain-specific individual scores (i.e., literature and visual arts domains) correlated with individual automatic scores (i.e., Word2Vec dissimilarity and AuDra score). Surprisingly, we found significant and positive correlations only between domains but not within domains (ICAA-literature and mean AuDra score, and ICAA-visual arts and mean Words2Vec dissimilarity, **Figure R1**).

These results indicate that individuals with higher creative engagement in the literature domain tend to achieve higher AuDra scores (reflecting their creativity during the Drawing task). Similarly, a positive relationship is observed between creative engagement in the visual arts domain and the Word2Vec mean dissimilarity score (which reflects creativity during the Word task).

These additional findings support a domain-general effect of expertise, indicating that creative achievements (i.e., those linked with domain-specific knowledge and expertise) are positively associated with creative performance beyond the boundaries of their domains of expertise. However, as these results fall outside the scope of our study, which focuses on evaluative mechanisms, we have decided not to include them in the manuscript. Nevertheless, we are open to including them in the Supplementary Results if the referees deem it relevant.

Figure R1 | Correlations (two-tailed, Spearman) between ICAA domain-specific scores and average automatic scores. One dot represents one participant.

- Results: the authors used SemDis to analyze the AUT. However, the correlation of semdis to human ratings is not as strong as more recent approaches, such as LLMs trained to predict human ratings - did the authors consider using this approach as it is more robust? Even though responses are in French, they can be scored by LLMs (<https://openscoring.du.edu/scoringllm>). This may explain relatively lower correlations between model parameters for alpha (cf. supplemental fig 2). Organisciak, P., Acar, S., Dumas, D., & Berthiaume, K. (2023). Beyond semantic distance: Automated scoring of divergent thinking greatly improves with large language models. *Thinking Skills and Creativity*, 49, 101356.

We thank the reviewer for raising this point and mentioning this recent approach. To address this concern, we conducted a scoring of responses from the Object domain (i.e., responses from the adapted version of the Alternate Uses Tasks) using OSCAI (Open Creativity Scoring with Artificial Intelligence). Interestingly, the correlation (two-tailed, Spearman) between SemDis and both OSCAI Originality and Elaboration scores were positive and significant (OSCAI_{Originality}: $\rho=0.20$, $p<1.10^{-3}$; OSCAI_{Elaboration}: $\rho=0.47$, $p=1.10^{-3}$, **Fig. R2**) suggesting a relative equivalence between the two scoring methods.

Next, we double-checked that the results reported with SemDis remain valid when using the OSCAI scores. First, the significant difference of OSCAI scores between *First* and *Creative* conditions at the group level was maintained (*Creative* vs. *First*: OSCAI_{Originality}: $t(72)=28.19$, $p=1.10^{-40}$; OSCAI_{Elaboration}: $t(72)=6.48$, $p=1.10^{-8}$).

Then, a significant canonical correlation was observed between creativity scores from the questionnaires and creativity scores from the tasks (replacing SemDis with OSCAI originality and elaboration scores) ($r=0.48$, $p=0.04$, **Supplementary Figure 5**). The contributions of each score to their associated canonical variable remained qualitatively and quantitatively similar. In particular, the contribution of OSCAI scores (both originality and elaboration) was still low and non-significant (Uses – Mean $OCSAI_{Originality}$ $r=0.04$, $p=0.76$; $OCSAI_{Elaboration}$: $r=0.02$, $p = 0.14$), consistent with the previous contribution of SemDis score.

Figure R2 | Two-tailed Spearman correlation between individual's mean SemDis and OSCAI scores.

Thus, in the absence of any significant changes in the results or their interpretation when using OSCAI scoring, we have decided to keep the use of SemDis scoring. However, we have included these results in the Supplementary Materials and have indicated in the Results and Methods sections that this control analysis has been conducted.

Results (L217 to 219)

Control analyses with alternative scoring method of Uses revealed similar results (see Supplementary Result 3) and AuDra scores were consistent with expert human ratings (Supplementary Result 4)

Results (L369 to 371)

Control analyses using an alternative scoring method of participants' responses in the Uses domains are detailed in Supplementary Results 3 and Supplementary Figure 5

(Methods – L725 to 730)

For the Uses task, we conducted control analyses using a recent alternative method for automatic scoring (Open Creativity Scoring with Artificial Intelligence, OSCAI)⁶⁵. All results presented with SemDis scores were replicated (including non-significance) with OSCAI scores (Supplementary Result 3 and Supplementary Figure 5). Additionally, we validated the reliability of AuDra scores by comparing them with human expert ratings. We found good reliability for scoring drawings with AuDra. (Supplementary Result 4)

We have included Supplementary Result 3 and Supplementary Figure 5, which summarize the results reported here.

(Supplementary – L864 to 886)

Supplementary Result 3 | Uses domain – Replication of results with an alternative scoring method.

We conducted a control analysis by using an alternative scoring method (Open Creativity Scoring with Artificial Intelligence, OSCAI)⁶⁵, which provide two scores per response: originality and elaboration ($OCSAI_{Originality}$ and $OCSAI_{Elaboration}$). We found a significant correlation between SemDis and OSCAI scores ($r(OCSAI_{Originality}, SemDis)=0.20$, $p=6.10^{-72}$; $r(OCSAI_{Elaboration}, SemDis)=0.47$, $p<1.10^{-6}$), with qualitatively and quantitatively similar results from the analyses conducted with SemDis scores (Supplementary Figure 5). The results indicate that participants' responses were more original and more elaborate in the *Creative* condition compared to the *First* condition (Uses

(OCSAI_{Originality}): Prediction_{First}=1.151±0.009, Prediction_{Creative}=2.441±0.044, t(72)=28.18, p=1.10⁻⁴⁰; Uses (OCSAI_{Elaboration}): Prediction_{First}=2.920±0.162, Prediction_{Creative}=4.021±0.188, t(72)=6.48, p=1.10⁻⁸. Then, regarding the canonical correlation analysis, the first canonical variable raised a significant correlation (r=0.48 p=0.048). All variables of the Questionnaire Set contributed significantly to its associated canonical variable (Questionnaire set: r_{CreaAch}=0.95, p=1.10⁻³⁵, r_{CreaAct}=0.63, p=4.10⁻⁹, r_{CreaSelf-report}=0.73, p=4.10⁻¹³, Pearson correlation coefficient between variables and canonical variable of the set). In the Model & Behavior set, the model's parameters (r_α=0.24, p=0.038, r_δ=-0.32, p=0.007) contributed significantly to their associated canonical variable. Regarding automatic scoring of creativity, both the AuDra score (r_{Drawings - AuDra}=0.81, p=7.10⁻¹⁸) and the Word2Vec score (negative cosine similarity) (r_{Words - Word2Vec}=0.64, p=1.10⁻⁹) contributed significantly and positively to the associated canonical variable. OCSAI scores did not show any significant contribution (r_{Uses - OCSAI (Originality)}=0.04, p=0.763; r_{Uses - OCSAI (Elaboration)}=0.18, p=0.137).

Supplementary Figure 5 | Canonical correlation between Model & Behavior set and Questionnaire set – OCSAI scoring for the Uses domain. (A) Correlation between the first canonical variable of the Questionnaire set and the first canonical variables of the Model & Behavior set. (B) Bar plots represent the correlation coefficients between each variable of the Model & Behavior set with the first canonical variable from the Model & Behavior set. For Words, Uses, and Drawings, variables are individual means of respective scores. (C) Same as (B) for each variable of the Questionnaire set with the first canonical variable from the Questionnaire set. Stars indicate the significance of the correlation (p<0.05). (Zoom panel) Spearman correlation between mean OCSAI originality and elaboration scores with SemDis score across participants.

Additionally, we updated the discussion as follows:

(Discussion – L546 to 556)

Finally, another limitation of the present protocol concerns the lack of a significant contribution of the SemDis score associated with Uses responses to the canonical variable related to the task (i.e. Model & Behavior set). The AUT typically involves producing several alternate uses of the same object within several minutes, depending on the version⁶⁴. In our task inspired by the AUT, however, the maximum time allowed in the Creative condition was limited to 20 seconds, and participants were instructed to provide only a single response. These modifications may have reduced the opportunity for participants to produce responses with sufficient variability in creativity, potentially affecting the observed contribution of the SemDis scores. Note that alternative automatic scoring methods such as OCSAI⁶⁵ yielded similar non-significant results (see Supplementary Result 3).

- Also, I may have missed it but I didn't catch zero-order correlations between originality ratings between the three tasks?

This question cannot be addressed at the trial level because the items were not identical across tasks (e.g., the cue “cow” in the Words domain had no equivalent in the Drawings or Uses domains). However, at the subject level, we can compute the mean originality ratings provided by each individual in each domain and examine inter-individual correlations.

The results reveal significant correlations between originality ratings across domains (see **Fig. R3**). These correlations can be interpreted in two ways:

- 1) Consistency in the use of the rating scale: Participants may have used the originality rating scale similarly across the three domains. For instance, some participants might consistently favor the lower end of the scale, while others might use the higher end. Alternatively, participants who used the entire scale might still consistently overestimate (or underestimate) originality across all three domains, while still using the whole scale.

- 2) Domain-generality of originality performance: Participants' responses might reach similar levels of originality in the three domains, suggesting domain-general performance in terms of originality.

Figure R3 | Pearson's correlations of originality ratings between domains at the subject level. One dot represents one participant.

To rule out the first interpretation, we examined correlations between automatic scores between domains instead of relying on subjective originality ratings. This analysis revealed only trends or non-significant correlations between automatic scores ($r(\text{Prediction}_{\text{Words}}, \text{Prediction}_{\text{Uses}}) = 0.09$, $p = 0.456$; $r(\text{Prediction}_{\text{Words}}, \text{Prediction}_{\text{Drawings}}) = 0.20$, $p = 0.085$; $r(\text{Prediction}_{\text{Drawings}}, \text{Prediction}_{\text{Uses}}) = 0.20$, $p = 0.092$, **Fig. R4**). These results do not provide sufficient evidence to eliminate the first interpretation or to conclude whether creative performance is domain-general or domain-specific. Furthermore, as automatic scores might reflect both originality and adequacy dimensions, and because we do not have objective measures of these two dimensions, any conclusions drawn from these results are inherently limited.

Figure R4 | Pearson's correlation between automatic scores between domains at the subject level. One dot represents one participant.

Altogether, we have decided not to include the results of these analyses in the manuscript or the Supplementary Results as they are not directly relevant to our main research question, which aimed to determine whether the integration of subjective adequacy and originality judgments (which may be accurate or biased) into a subjective value is consistent across domains. However, if the reviewer believes these correlations are important, we are open to including them in the Supplementary Material and would welcome guidance on the added value of this integration.

5. Figure 1 was a lot to process. I appreciate the effort but I wonder if this could be made more simple/easier to digest?

We thank the reviewer for pointing that out. We addressed it by changing the design of the schematic representation of the tasks, to make it clearer:

Figure 1 | Schematic representation of the experimental design. (A) Task order. Participants performed the First condition of the Free Generation tasks. Then, they completed the Creative condition of these tasks. The generation tasks were then followed by the likeability rating tasks. Next, they completed the adequacy and originality rating tasks. Finally, they fulfilled a set of questionnaires (ICAA and self-report). (B) Detail of the tasks for each domain: Words, Uses, and Drawings. Within each domain, the order of tasks was counterbalanced within and between participants. In the First condition (first column), participants were asked to provide the first response (word, use, or drawing) that came to mind when presented with a cue. In the Creative condition (second column), they were instructed to provide creative responses to the same cues. Then (third column), participants saw their responses along with other potential responses and rated how much they liked each association (symbolized with a heart). Finally (fourth column), they rated the same associations for adequacy (symbolized with a target) and originality (symbolized with a star). The order of these ratings was randomized for each association. Participants' actions (keyboard pressing or mouse clicking) are symbolized with hands and mouse computer, and delimit timing variables (i.e., response and production times) that elapsed through the tests are symbolized vertically.

6. Figure 2 is labeled “Motivation Effect” but the text does not really refer to it as such. In fact, the term “motivation” doesn’t appear until the Discussion, so I would discuss it sooner (though I know “energizing” has been used earlier). This could make things more clear.

To address this comment, we revised the introduction, to emphasize earlier the relationship between the present work and motivational processes.

(Introduction – L87 to 101)

Our previous work²¹ has provided a framework that allows to dissociate evaluative processes from generative processes during a creativity task and decomposes evaluation into three specific subprocesses: (i) the monitoring of ideas’ originality and their adequacy to the context, (ii) the valuation of the ideas, i.e. the assignment of a subjective value based on originality and adequacy, according to individual preferences, and (iii) the selection of the idea with the highest subjective value. This framework was tested in an experimental design that combined a semantic creativity task and participants’ self-judgments of their responses in terms of originality, adequacy, and subjective values. Model fitting and simulations showed that subjective value drove the selection of an idea among potential candidates, in three main findings. First, we found that, during the creativity task, preferred ideas were provided faster than less valued ideas. These results suggest that the likeability of an idea energizes its production, emphasizing the potential underlying motivational role of valuation in creative idea production²². Indeed, valuation leads to behavioral energization²³, meaning that subjective values predict how much effort is put into an action, or in our case, how fast an action is performed, reflecting motivation.

Additionally, we harmonized the different terms associated with motivation, to make our message clearer, specifically when proposing our hypothesis.

(Introduction – L130 to 134)

Our specific hypotheses were that (i) creative idea production is energized by valuation in the three creative domains (supporting the domain-general role of motivation), (ii) subjective values are built on both the adequacy and originality of response, and similarly across the three domains, and (iii) individual differences in preferences are related to creative abilities in real-life.

7. Figure 4, I suggest making panel C y axis more granular than just 0, .5, 1.

We have updated Figure 4. In this new version, we added granularity to panel C's y-axis and panel B's x-axis.

Figure 4 | Relevance of model parameters and behavioral variables regarding creative abilities. (A) Correlation between the first canonical variable of the Questionnaire set and the first canonical variable of the Model & Behavior set. (B) Bar plots represent the correlation coefficient between each variable of the Model & Behavior set with the first canonical variable from the Model & Behavior set. For Words, Uses, and Drawings, variables are individual means of respective scores. (C) Same as (B) for each variable of the Questionnaire set with the first canonical variable from the Questionnaire set. Stars indicate the significance of the correlation ($p < 0.05$).

8. I think the Discussion would also benefit from mentioning the potential role of expertise, as noted earlier, beyond lab tasks to real-world tasks requiring expert knowledge and skills.

We thank the reviewer for their comment and now propose a development on this topic in the discussion section.

(Discussion – L502 to 513)

Domain expertise, which is closely linked to memory and past experience, may influence the way adequacy and originality are monitored and assessed. For instance, experts might possess a richer set of reference points against which they can assess the originality and adequacy of ideas within their field. Consequently, experts might more accurately assess adequacy and originality, potentially enhancing their creative performance within their field. Recent experimental findings indicate that experts outperform novices in creative tasks directly related to their field of expertise (e.g., music improvisation⁵⁵), yet, no significant differences between novices and experts were observed on classic divergent thinking tasks, such as the AUT^{11,56}. Thus, expertise and domain knowledge are better predictors of domain-specific creativity⁵⁷, but it is unclear whether this advantage arises from better idea generation, better monitoring, or both. The present study did not formally address the relationship between domain expertise and creativity.

Moreover, we integrated perspectives regarding future investigations of this point:

(Discussion – L513 to 516)

A promising direction for future studies would be to test the domain-generality of valuation in creative experts across different domains to determine whether expertise influences how creative ideas are liked and how their originality and adequacy are assessed.

Additionally, as the present comment echoes the comment #2, we refer the reviewer to their previous comments for which we provided an integration of this point in the introduction section.

9. Can the authors explain why they used drawing stimuli that were not in the original training set of AuDrA? The paper shows that the model does not perform well on drawing stimuli that it was not trained on. I would also like to see some validation evidence to show the drawing scores worked as expected; I suspect the values may be driven by pixel count/elaboration. Also, why could participants not erase what they drew? <https://link.springer.com/article/10.3758/s13428-023-02258-3>

The reviewer raises an interesting point here and we thank them for that. Our initial choice was to adapt the stimuli from Nishimoto et al., (2010) as we needed 30 different stimuli to match the number of trials in the other domains (the training set of AuDra contains only 12 stimuli).

Also, we decided not to allow participants to erase what they drew to capture the first idea that came to their minds in the *First* condition and force them to think before starting to draw in the *Creative* condition. This also simplified the computation of drawing speed. We have now included these justifications in the Methods section.

(Methods – L617 to 620)

All cues are available in Supplementary Method 1 and 2 and Supplementary Figure 4. Participants used the computer mouse to draw. Erasing was not allowed to capture the first idea that came to mind in the *First* condition, encourage participants to think before starting to draw in the *Creative* condition, and to simplify the computation of drawing speed.

The original work of Patterson et al. (2024) indeed showed that AuDra performs better with images from the MTCTI (Incomplete Shape Task, Barbot et al. 2018). However, the relationship between AuDra predictions and human ratings for other types of figural creativity tasks showed a reasonable correlation ($r=0.49$, $MSE=0.034$).

The first evidence of an accurate creativity scoring by AuDra is the result we report in our manuscript, which shows higher AuDra scores for the Drawings *Creative* condition compared to the *First* condition ($t(72)=-10.07$, $p=2.10^{-15}$, **Figure R5, Panel A**).

Regarding the pixel count/elaboration, the original work of Patterson et al. showed that, regardless of the generalization of AuDra (i.e., new rater, new drawings, or new task), the correlation between AuDra scores and human ratings consistently outperforms the correlation between elaboration metrics (i.e., number of inked pixels) and human ratings. Nevertheless, we checked for the potential presence of elaboration effects in our dataset. We first examined the correlation between AuDra scores and drawing length across all responses and found a significant effect ($r=0.56$, $p<1.10^{-3}$), confirming that AuDra scores are indeed partly related to pixel count or elaboration (**Figure R5, Panel B**).

To determine whether the initial difference in AuDra scores between responses from the *First* and *Creative* conditions in the Drawings task could be fully accounted for by the drawing length, we orthogonalized AuDra scores by drawing length (AuDra*) and compared AuDra* scores between

the *First* and *Creative* conditions. We found that the difference remained significant ($t(72)=-3.1031$, $p=0.0027$, **Figure R5, Panel C**), suggesting that AuDra captures more than just pixel count or elaboration.

Figure R5 | Validation of AuDra scores – Relationship with elaboration. (A) Mean individual AuDra score per experimental condition. (B) Pearson's correlation between AuDra scores and drawing's length. (C) Mean individual AuDra* score (AuDra score orthogonalized for response length) per experimental condition.

Finally, we double-checked that AuDra applied to our drawings was working as expected by comparing AuDra scores and human creativity ratings to a subset of our drawings (120 out of 4500). Spearman correlation between AuDra scores and mean human ratings was significant and illustrated in **Figure R6**.

Figure R6 | Two-tailed Spearman correlation between AuDra scores and mean human ratings for a selection of 120 drawings from our dataset.

Altogether, we conclude that despite the effective correlation between pixel count and AuDra score, the latter still provides a reasonable estimation of creativity. We integrated the following paragraph in the Supplementary Methods section:

(Supplementary – L888 to 905)

Supplementary Result 4 | Drawing domain – Estimation of AuDra reliability with human expert ratings.

To confirm the reliability of AuDra scores, we used the consensual assessment technique recommended for evaluating creativity tasks⁵². For each of the 30 abstract shapes used as cues, we selected four participants' drawings based on the quantile distribution of the AuDra score for those abstract shapes. This method ensures a balanced diversity of AuDra scores for each cue. Next, seven judges from the lab, all experienced in creativity assessments but unfamiliar with the data, were recruited. The judges first viewed the 30 abstract shapes, each followed by the four selected drawings. After this initial viewing, the 120 selected drawings were presented in a randomized order, and judges rated them on a scale from 0 (not creative at all) to 4 (extremely creative). Intraclass correlation estimates indicated good reliability among judges' ratings (ICC=0.858, CI= [0.816 0.893]), with 95% confidence intervals, based on a mean-rating (k=7), absolute agreement, and 2-way random-effects model. A Spearman correlation analysis between the mean of the judges' ratings and the AuDra scores revealed a significant positive relationship ($r=0.47$, $p<1.10^{-3}$). Note that in Patterson et al (2024)³³ the correlation between AuDra scores and human raters for drawings that do not include the original abstract shapes on which the model was trained yielded similar results ($r=0.49$, 95% CI [.43, .54]).

And refer to this supplementary section in the results:

Results (L217 to 219)

Control analyses with alternative scoring method of Uses revealed similar results (see Supplementary Result 3) and AuDra scores were consistent with expert human ratings (Supplementary Result 4)

Reviewer #2

The present manuscript, 'Subjective valuation as a domain-general process in creative thinking' addresses the question of whether or not creative valuation is a domain general process. Findings suggest a consistent valuation mechanism across word association, object uses and drawing tasks. Authors make the case that creative evaluation is inherently domain general. While the topic is of general interest and the study is well presented, more work is needed to provide background on previous research in this area and **soften the claims of domain generality**. Additionally, there are several instances throughout the manuscript where **language could be improved for clarity and precision**. See below for specific comments:

We thank the reviewer for their careful assessment of our manuscript. We have revised our manuscript to provide more background on previous research and double-checked that the claim of domain-generality was only associated with valuation (as we agree that judgments of originality and adequacy could differ between domains) and improve language clarity and precision.

1. - On p. 3, line 51, authors cite findings from Reiter-Palmon and colleauges (2009) to support that divergent thinking tasks have predictive validity of real-world creative achievement. What about studies looking at **DT performance in creative experts**? There is more work on this topic which should be referenced here, expanding on the domain-generality of divergent thinking in real-world creative people.

We thank the reviewer for raising this point and decided to group this comment with Comments #2 and #8 from Reviewer 1 on the role of expertise and its relevance for creativity in specific domains. We now address the topic of expertise in the introduction and the discussion of the manuscript:

Introduction (L56 to 65)

For instance, scores of creative achievements and activities in the literature domain tend to correlate with both achievements and activities in a wide range of domains, such as culinary and visual arts, but the highest correlation remains for creative writing itself. These findings from questionnaires are consistent with empirical evidence⁹ indicating that life experiences, such as education and cultural background, have a stronger impact on domain-specific creativity than on domain-general creativity. Overall, these findings suggest that creativity has both domain-general and domain-specific mechanisms, the latter potentially developed from previous experiences and domain expertise^{10,11}. A key remaining question is the extent to which the core cognitive mechanisms underlying creative thinking are domain-general or domain-specific.

(Discussion – L502 to 513)

Domain expertise, which is closely linked to memory and past experience, may influence the way adequacy and originality are monitored and assessed. For instance, experts might possess a richer set of reference points against which they can assess the originality and adequacy of ideas within their field. Consequently, experts might more accurately assess adequacy and originality, potentially enhancing their creative performance within their field. Recent experimental findings indicate that experts outperform novices in creative tasks directly related to their field of expertise (e.g. music improvisation⁵⁵), yet, no significant differences between novices and experts were observed on classic divergent thinking tasks, such as the AUT^{11,56}. Thus, expertise and domain knowledge are better predictors of domain-specific creativity⁵⁷, but it is unclear whether this advantage arises from better idea generation, better monitoring, or both. The present study did not formally address the relationship between domain expertise and creativity

(Discussion – L513 to 516)

A promising direction for future studies would be to test the domain-generalty of valuation in creative experts across different domains to determine whether expertise influences how creative ideas are liked and how their originality and adequacy are assessed.

2. - Moreover, in line 73, authors outline the dual-process model of creativity. A recent paper by Benedek et al. (2023) provides a detailed model of this framework and the contributions of memory to the generative/evaluative stages of creativity. The introduction of the present manuscript would be improved by a brief discussion of this framework, as a model for the creative process.

This point raised by the reviewer offers an additional perspective to the present study. Therefore, we added the following section to the discussion of the manuscript (to maintain the focus on valuation in the introduction, we decided to discuss this framework in the discussion instead):

(Discussion– L493 to 502)

A recent framework, consistent with the two-stage view of creativity, proposes that episodic and semantic memory contributes to the generation but also the evaluation stage⁵¹. This implies that domain-specific knowledge can influence both the generation and evaluation phases of creativity. Our findings are not inconsistent with this view. Here, we do not claim that judgments of originality and adequacy are inherently domain-general⁵²⁻⁵⁴. Rather, we argue that the process by which originality and adequacy are integrated – regardless of their accuracy- is domain-general. In other words, our results support the hypothesis that the mechanism governing this integration is consistent across domains. However, these results do not exclude the potential influence of memory and past experiences in a specific domain on other processes of the evaluation stage.

3. - The schematic representation in Figure 1 is overly complex and difficult to interpret. I recommend simplifying this figure to be more readily interpretable.

We updated the schematic representation of the experimental design to make it clearer and easier to understand:

Figure 1 | Schematic representation of the experimental design. (A) Task order. Participants performed the *First* condition of the Free Generation tasks. Then, they completed the *Creative* condition of these tasks. The generation tasks were then followed by the likeability rating tasks. Next, they completed the adequacy and originality rating tasks. Finally, they fulfilled a set of questionnaires (ICAA and self-report). **(B) Detail of the tasks for each domain: Words, Uses, and Drawings.** Within each domain, the order of tasks was counterbalanced within and between participants. In the *First* condition (first column), participants were asked to provide the first response (word, use, or drawing) that came to mind when presented with a cue. In the *Creative* condition (second column), they were instructed to provide creative responses to the same cues. Then (third column), participants saw their responses along with other potential responses and rated how much they liked each association (symbolized with a heart). Finally (fourth column), they rated the same associations for adequacy (symbolized with a target) and originality (symbolized with a star). The order of these ratings was randomized for each association. Participants' actions (keyboard pressing or mouse clicking) are symbolized with hands and mouse computer, and delimit timing variables (i.e., response and production times) that elapsed through the tests are symbolized vertically.

4. - In the results, authors report that participants rated responses from the creative condition as being more original than the first responses. Could it be the case that there is a bias in how participants rate their responses in the two different conditions. In other words, that the responses do not vary in their originality, but there is some bias in how participants are ratings their own responses across these conditions?

We thank the reviewer for highlighting this relevant point. Our design allows us to test for such a bias, as participants rated both their own productions and pre-defined productions (attributed to *First* and *Creative* conditions, see Methods). We conducted a mixed-model analysis on originality ratings with two factors: the source of the association (self vs other) and the condition during which the association was produced (*First* or *Creative*), with subjects and domains as random factors.

Predictor	DF	Estimate	SE	T-Statistic	P-Value
(Intercept)	21771	68.074	3.61	18.84	2.10⁻⁷⁸
Source	21771	-4.57	0.54	-8.54	1.10⁻¹⁷
Condition	21771	-28.88	0.55	-52.05	<1.10⁻⁶
Source*Condition	21771	-2.578	0.75	-3.43	6.10⁻⁴
Formula	Originality ~ 1 + isSub*RatingCond + (1 ID) + (1 Domain)				

Table R1 | Summary of model statistics. Results of mixed-model analysis regarding the effect of the source, and the condition.

The results indicate a significant effect of the condition ($t(21771)=-28.88\pm 0.55$, $p<1.10^{-6}$), showing that responses from the *First* condition were rated as less original compared to those from the *Creative* condition.

Additionally, there was a significant effect of the source ($t(21771)=-4.57\pm 0.54$, $p=1.10^{-17}$), indicating that participants rated their own responses as less original than other responses.

Finally, the interaction term was significant ($t(21771)=-2.58\pm 0.75$, $p=6.10^{-4}$), indicating that the difference in originality ratings between *First* and *Creative* responses was higher for participants' own responses than for other responses (**Fig. R7**).

Altogether, these results indicate that, as reported in the manuscript, participants rated their responses from the *Creative* condition as more original than their responses from the *First* condition. The interaction shows that the difference between *First* and *Creative* was stronger for their own responses than for other responses, suggesting the presence of a bias. However, the effect of condition was also observed for other responses, indicating that participants' originality judgments were consistent across sources.

Figure R7 | Average originality ratings for responses from the First and Creative conditions from subjects and others. Statistical significance is indicated by stars.

Moreover, the fact that participants judged other responses as more original than their own suggests that they might be more “severe” with themselves than with others. Note that the size effect of the condition is much larger than that for the source or the interaction, suggesting that the variance in originality ratings is mainly explained by the condition and only to a lesser extent by the source.

Furthermore, we reported that the automatic “creativity” scores were higher for participants’ *Creative* responses than for their *First* responses, supporting the finding that their responses were indeed more original in the *Creative* than in the *First* condition.

In summary, we believe that this bias does not undermine the conclusions of our results, which focus on how originality and adequacy are integrated into a subjective value. If the reviewer considers this bias problematic, we would be glad to address it further with their guidance.

5. - On p. 8, line 179, authors state, "this result is consistent with the fact that producing creative responses takes more time." A citation is needed here to support this claim.

We thank the reviewer for spotting this omission. We added two references to this statement:

(Discussion L406 to 408):

Resource allocation is especially relevant regarding the effortful process that is creativity³⁰, where achieving higher originality requires more effort²⁹

Attached references to the previous section:

29. Barbot, B. *The Dynamics of Creative Ideation: Introducing a New Assessment Paradigm*. *Front. Psychol.* 9, (2018).

30. Beaty, R. E. & Silvia, P. J. *Why do ideas get more creative across time? An executive interpretation of the serial order effect in divergent thinking tasks*. *Psychol. Aesthet. Creat. Arts* 6, 309–319 (2012).

6. - On line 184 authors describe the use of semantic distance as a proxy for creativity assessment. While these approaches are informative, the Open Creativity Scoring platform significantly outperforms these word2vec models and is generally accepted as a more reliable method for scoring originality of AUT responses. Is there a reason why the authors opted for semantic distance approaches over this Open Scoring method? If not, I strongly recommend repeating the analysis with this approach (see reference below). Organisciak, P., Dumas, D., Acar, S., and de Chantal, P. L. (2024). Open Creativity Scoring [Computer software]. Denver, CO: University of Denver. <https://openscoring.du.edu>.

We thank the reviewer for this suggestion. At the time the project started, SemDis was the most relevant method, to our knowledge, to provide an automatic scoring of our responses in the Object domains. Regarding the use of the open scoring method of Organisciak et al. (2024) and based on the comments of Reviewer #1 and #2, we included control analysis in the Supplementary Materials of the manuscript (we refer the reviewer to Reviewer #1 Comment #3 for further detail on this point). In a summary, SemDis and OCSAI scores share a high positive and significant relationship. With OCSAI scores instead of SemDis score in our analyses, we still observe a

significant difference in scores in the *Creative* condition compared to the *First* condition (OCSAI_{Originality}: $\rho=0.20$, $p<1.10^{-3}$; OCSAI_{Elaboration}: $\rho=0.47$, $p=1.10^{-3}$, **Fig. R2**). Additionally, the integration of OCSAI scores in the canonical correlation analysis does not change the results qualitatively or quantitatively. Thus, we decided to keep the analyses with SemDis score and provide the results with OCSAI scores in the Supplementary Materials, with a mention in the Results, Methods, and Discussion sections.

Reviewer #3

This manuscript describes a study investigating whether the evaluation process in creativity is domain-general or specific. They asked 73 participants to complete three different creativity tasks and evaluate their output, across three domains: verbal creativity, drawing and utilitarian or conceptual creativity. They also asked them to complete the ICAA – an inventory of creative activities and achievements. The authors hypothesized that (i) creative idea production is energized by valuation across domains, (ii) the construction of subjective values is based on both the adequacy and originality of response, also similarly across domains, (iii) preferences are stable across domains and (iv) correlated with creative abilities in real-life.

They found that valuation increased motivation across domains, in line with their first hypothesis. They also found that preferences are stable across domains and correlated with creative abilities in real-life. Finally, the authors found that valuation is indeed based on both originality and adequacy, regardless of domain.

In both semantic and utilitarian creativity, positive valuation was positively correlated with speed of responses – that is, the more one judged an item as valuable, the faster the response (or the shorter the response time). For drawing, however, preferred drawings tended to take longer to produce. This is an interesting distinction **that merits further study**, as effort and motivation might result in differential engagement in specific domains. Future directions should include neuroimaging as a way of better understanding what underlies these differences.

Overall, this paper is thorough and important, adequately powered, and carefully analyzed. More work is needed, however, to improve the **clarity and general understandability** of the arguments and results. I have made some notes below, but the authors should carefully re-read their manuscript to align tenses, and improve wording.

With these improvements, I believe this manuscript would make a positive addition to the literature and is worthy of publication.

We thank the reviewer for their careful and positive assessment of our manuscript. We have revised the manuscript to improve clarity and general understandability. Edits are highlighted in the revised version of our manuscript.

We have also added a sentence regarding the effect of motivation on different domains of creativity:

(Discussion – L402 to 411)

In summary, energization in verbal tasks and effort in drawing production can be seen as two sides of the same coin, both reflecting resource allocation. In some contexts, resource allocation is associated with an increased production speed (e.g., typing responses on a keyboard) or duration (e.g., drawing with a computer mouse). Resource allocation is especially relevant regarding the effortful process that is creativity³⁰, where achieving higher originality requires more effort²⁹. This mirrors findings from the effort allocation literature, where individuals exert more effort for greater rewards^{26,35}. Finally, these differential motivational effects in the figural domain highlight the need for further research, as motivation may drive different forms of effort or engagement depending on the creative domain.

Additionally, we improved the clarity of the discussion section relating to future neuroimaging perspectives:

(Discussion – L518 to 527)

At the neural level, future neuroimaging studies could investigate the neural representation of generation and valuation processes across different creative domains. A recent fMRI study²⁵ using the GMVS framework demonstrated the involvement of the Brain Valuation System (BVS), which encodes the likeability of ideas during their production. Based on our results and in the light of this literature, we propose two hypotheses: (i) the Brain Valuation System will represent the likeability of ideas across all domains, given its generic property⁵⁸, i.e., it represents likeability of any kind of item; and (ii) the exploration of the internal repertoire may differ between creative domains, depending on whether the creative output is semantic or figural. This domain-dependent exploration might be reflected in distinct brain mechanisms, supported by distinct brain regions^{15,20} or different activity patterns within a given network.

Page 1, line 14: 'It is admitted' change to Experts agree that

Line 16: "The evaluation phase involves valuation processes upstream selection" not sure what this means

Line 17: valuation of ideas not value?

Line 55: Perspectives in the field - is that a specific reference? Why is it past-tense?

Line 61: achievements 'tend' (remove 's')

Line 63: with a study...

Line 70: Remove 'a' before 'generation' and 'evaluation'

Line 78: align tenses

Line 79: 'The authors found' rather than 'it was'

Throughout that paragraph, align tenses.

Line 116: 'Alternate uses, not alternative', throughout the manuscript

Line 415: 'significantly better explains' is awkward

Line 417: add 'literature' or 'research' to 'past'

Line 465: remove 'intuitively'

We have corrected these reported issues and carefully rechecked and improved language in our manuscript (changes highlighted in yellow in the manuscript).